# Supported Value Regularization for Offline Reinforcement Learning

**Yixiu Mao**[1], **Hongchang Zhang**[1], **Chen Chen**[1], **Yi Xu**[2], **Xiangyang Ji**[1]
[1]Department of Automation, Tsinghua University
[2]School of Artificial Intelligence, Dalian University of Technology
`myx21@mails.tsinghua.edu, xyji@tsinghua.edu`

## Abstract

Offline reinforcement learning suffers from the extrapolation error and value overestimation caused by out-of-distribution (OOD) actions. To mitigate this issue, value regularization approaches aim to penalize the learned value functions to assign lower values to OOD actions. However, existing value regularization methods lack a proper distinction between the regularization effects on in-distribution (ID) and OOD actions, and fail to guarantee optimal convergence results of the policy. To this end, we propose Supported Value Regularization (SVR), which penalizes the $Q$-values for *all* OOD actions while maintaining *standard* Bellman updates for ID ones. Specifically, we utilize the bias of importance sampling to compute the summation of $Q$-values over the entire OOD region, which serves as the penalty for policy evaluation. This design automatically separates the regularization for ID and OOD actions without manually distinguishing between them. In tabular MDP, we show that the policy evaluation operator of SVR is a contraction, whose fixed point outputs unbiased $Q$-values for ID actions and underestimated $Q$-values for OOD actions. Furthermore, the policy iteration with SVR guarantees strict policy improvement until convergence to the optimal support-constrained policy in the dataset. Empirically, we validate the theoretical properties of SVR in a tabular maze environment and demonstrate its state-of-the-art performance on a range of continuous control tasks in the D4RL benchmark.

## 1 Introduction

Offline Reinforcement Learning (RL) aims to learn a policy from a fixed dataset collected by some behavior policy [24, 26]. It can tap into existing large-scale datasets [15, 30] for safe and efficient learning. However, it suffers from the extrapolation error [10] caused by out-of-distribution (OOD) actions, which can further result in severe overestimation of value functions [26].

To mitigate this issue, value regularization approaches attempt to penalize the $Q$-values of OOD actions to introduce conservatism in value estimation [23, 20, 46, 14, 3, 6]. However, we observe that existing value regularization methods not only fall short in penalizing all OOD $Q$-values but also may introduce detrimental changes to in-distribution (ID) ones. Specifically, most of them involve adding penalties to the critic learning objective [23, 50, 27, 6, 48]. However, due to the difficulty of distinguishing between ID and OOD actions, they typically adopt the idea of being pessimistic about the actions under the current policy and optionally optimistic about the ones within the dataset [23, 50, 46, 6]. Rather than relying on ID or OOD, this regularization is essentially based on the policy density, which we show is problematic when the dataset is heavily corrupted by sub-optimal actions. Other works involve reducing the Bellman target with uncertainty quantifiers [14, 3] or ensembles [2, 34]. However, if not incorporating additional penalties, they do not provide learning signals on OOD $Q$-values, logically only suppressing the overestimation in the ID region [14, 2].

In this work, we revisit the original objective of value regularization and ask a question - "Can we devise a value regularization method that penalizes *all* OOD Q-values *without* affecting ID ones"? We point out that the inability of prior methods to achieve this leads to a lack of strong theoretical guarantees for policy performance. In offline RL, the best policy that can be guaranteed should lie within the support of the behavior policy, known as the optimal support-constrained policy [22]. However, existing value regularization methods do not provide reliable guarantees for converging to it. On the other hand, several endeavors aim to learn it through specific policy constraints [10, 22, 11], but their empirical performance leaves considerable room for improvement.

To this end, we propose Supported Value Regularization (SVR), a new value regularization method that penalizes the Q-values for *all* OOD actions while maintaining *standard* Bellman updates for ID ones. Specifically, we leverage the bias of importance sampling to calculate the summation of Q-values over the entire OOD region, which serves as the penalty term for policy evaluation. This design circumvents the dilemma of distinguishing between ID and OOD actions and separates the regularization effects on them automatically. Theoretically, SVR offers stronger guarantees than previous methods. Under tabular MDP, we show that the SVR policy evaluation operator is a contraction in the whole state-action space and its fixed point outputs unbiased Q-values for ID actions and underestimated Q-values for OOD ones, while the fixed point of Q in CQL [23] may underestimate or overestimate Q-values. More importantly, SVR guarantees strict policy improvement until convergence to the optimal support-constrained policy, while prior value regularization methods lack such each-step improvement and global optimal convergence guarantees.

In practice, SVR is easy to implement by adding a penalty term to the ordinary policy evaluation loss. Empirically, we validate the support-constrained optimality of SVR in a tabular maze environment, where baselines fail to converge. Moreover, SVR achieves state-of-the-art performance on a range of continuous control tasks in the D4RL benchmark [7] and shows strong advantages on noisy datasets.

## 2 Preliminaries

**Offline RL.** The environment in RL is typically modeled as a Markov Decision Process (MDP) $\mathcal{M} = (\mathcal{S}, \mathcal{A}, P, R, \gamma, d_0)$, with state space $\mathcal{S}$, action space $\mathcal{A}$, transition dynamics $P : \mathcal{S} \times \mathcal{A} \to \Delta(\mathcal{S})$, reward function $R : \mathcal{S} \times \mathcal{A} \to [R_{\min}, R_{\max}]$, discount factor $\gamma \in [0, 1)$, and initial state distribution $d_0$ [41]. The goal of RL is to find a policy $\pi : \mathcal{S} \to \Delta(\mathcal{A})$ that maximizes the expected discounted return: $\mathbb{E}_{s_0 \sim d_0, a_t \sim \pi(\cdot|s_t), s_{t+1} \sim P(\cdot|s_t, a_t)}[\sum_{t=0}^{\infty} \gamma^t R(s_t, a_t)]$. For any policy $\pi$, we define the value function as $V^\pi(s) = \mathbb{E}_\pi[\sum_{t=0}^{\infty} \gamma^t R(s_t, a_t)|s_0 = s]$ and the state-action value function (Q-value function) as $Q^\pi(s, a) = \mathbb{E}_\pi[\sum_{t=0}^{\infty} \gamma^t R(s_t, a_t)|s_0 = s, a_0 = a]$. By the boundedness of rewards, we have $Q^\pi \in [Q_{\min}, Q_{\max}]$, where $Q_{\min} := R_{\min}/(1-\gamma)$ and $Q_{\max} := R_{\max}/(1-\gamma)$. In addition, we use $\rho_{sa}^\pi$ to denote the normalized and discounted state-action occupancy of policy $\pi$ with the initial state-action pair $(s, a)$: $\rho_{sa}^\pi(s', a') = (1-\gamma)\sum_{t=0}^{\infty} \gamma^t \mathbb{E}_\pi[\mathbb{I}[s_t = s', a_t = a']|s_0 = s, a_0 = a]$.

Actor-critic methods [19] alternate between evaluating the policy by iterating the Bellman operator $(\mathcal{T}^\pi Q)(s, a) := R(s, a) + \gamma \mathbb{E}_{s'} \mathbb{E}_{a' \sim \pi(\cdot|s')}[Q(s', a')]$, and improving the policy by maximizing the Q-value. In offline RL, the agent is provided with a fixed dataset $\mathcal{D}$ collected by some behavior policy $\beta$. Ordinary actor-critic algorithms [9, 12, 39] minimize the following losses alternately:

$$L_Q(\theta) = \mathbb{E}_{(s,a,s') \sim \mathcal{D}}[(Q_\theta(s, a) - R(s, a) - \gamma \mathbb{E}_{a' \sim \pi_\phi(\cdot|s')} Q_{\theta'}(s', a'))^2] \quad \text{(policy evaluation)} \quad (1)$$

$$L_\pi(\phi) = -\mathbb{E}_{s \sim \mathcal{D}, a \sim \pi_\phi}[Q_\theta(s, a)] \quad \text{(policy improvement)} \quad (2)$$

where $\pi_\phi$ is a policy parameterized by $\phi$, $Q_\theta(s, a)$ is a Q function parameterized by $\theta$, and $Q_{\theta'}(s, a)$ is a target Q function whose parameters are updated via Polyak averaging [32].

**Value regularization.** In offline RL, OOD actions $a'$ can produce erroneous Bellman targets and lead to an inaccurate estimation of Q-values. Then in policy improvement, the policy tends to prioritize OOD actions whose values have been overestimated, resulting in poor performance.

To address this issue, value regularization methods regularize the Q function to introduce conservatism in value estimation [23, 20]. As the most representative one, CQL [23] minimizes the following policy evaluation loss, which guarantees to obtain underestimated $V$ functions:

$$\min_Q \mathbb{E}_{s \sim \mathcal{D}, a \sim \beta}\left[(Q(s, a) - \mathcal{T}^\pi Q'(s, a))^2\right] + \alpha \left(\mathbb{E}_{s \sim \mathcal{D}, a \sim \pi}[Q(s, a)] - \mathbb{E}_{s \sim \mathcal{D}, a \sim \beta}[Q(s, a)]\right) \quad (3)$$

where $Q'$ is the target Q function and $\alpha$ is a hyperparameter.

# 3 Supported Value Regularization

In this section, we first briefly analyze the existing density-based value regularization and identify some important issues. Then we propose Supported Value Regularization (SVR) to penalize all OOD $Q$-values while maintaining standard updates for ID ones. Next, we conduct a thorough analysis of SVR and demonstrate its theoretical superiority. Last, we present the implementation details of SVR.

## 3.1 Density-based value regularization

A large portion of value regularization methods are built on the idea of being pessimistic about the actions under current policy and optimistic about the actions within the dataset [23, 50, 27, 6]. Instead of relying on ID or OOD, we show that this regularization is essentially based on the policy density, which we refer to as density-based value regularization. Specifically, we take CQL (Eq. (3)) [23], the foundational work on value regularization, as an example for mathematical analysis. Here we remove the strong assumption $\mathrm{supp}(\pi) \subseteq \mathrm{supp}(\beta)$ in their paper and analyze its regularization effects on the $Q$ functions. From Eq. (3), we can obtain the analytical solution of $Q$, which corresponds to the following policy evaluation operator $\mathcal{T}_{\mathrm{CQL}}^{\pi}$:

$$
\mathcal{T}_{\mathrm{CQL}}^{\pi} Q(s, a) = \begin{cases} \mathcal{T}^{\pi} Q(s, a) - \alpha \left( \frac{\pi(a|s)}{\beta(a|s)} - 1 \right), & \beta(a|s) > 0, \\ -\infty, & \beta(a|s) = 0, \pi(a|s) > 0, \\ Q(s, a), & \beta(a|s) = 0, \pi(a|s) = 0. \end{cases}
\tag{4}
$$

Considering the iteration of policy evaluation: $Q^{k+1}(s, a) = \mathcal{T}_{\mathrm{CQL}}^{\pi} Q^k(s, a)$, we have Observation 1:

**Observation 1.** *In each iteration $k$, compared to $\mathcal{T}^{\pi}$, $\mathcal{T}_{\mathrm{CQL}}^{\pi}$ (1) lowers $Q^{k+1}(s, a)$ when $\pi(a|s) > \beta(a|s)$; (2) raises $Q^{k+1}(s, a)$ when $\pi(a|s) < \beta(a|s)$; (3) obtains the same $Q^{k+1}(s, a)$ when $\pi(a|s) = \beta(a|s) > 0$; (4) does not update $Q^{k+1}(s, a)$ when $\pi(a|s) = \beta(a|s) = 0$.*

For (1)(2)(3), lowering or raising $Q$-values based on the relative density between $\pi$ and $\beta$ could be problematic, especially when the dataset contains a large portion of bad actions. In such cases, assuming the policy $\pi$ has found the optimal action $a^*$, it holds that $\pi(a^*|s) > \beta(a^*|s)$, and thus CQL will lower $Q(s, a^*)$; for some bad action $\hat{a} \in \mathcal{D}$, it holds that $\pi(\hat{a}|s) < \beta(\hat{a}|s)$, and thus CQL will raise $Q(s, \hat{a})$. That is, CQL tends to raise the $Q$-values of numerous bad actions and lower the $Q$-values of scare good actions, forcing the policy to choose bad actions in policy improvement stage. Regarding (4), in the entire OOD region ($\beta(a|s) = 0$), the penalization is only performed where $\pi(a|s) > 0$. Since the distribution of $\pi$ is narrow compared to the whole action space, the regularization region of CQL and most existing methods [50, 6, 27, 3, 48] tends to be too narrow.

## 3.2 Supported value regularization

In this paper, we aim to penalize $Q$-values for all OOD actions ($a \notin \mathrm{supp}(\beta)$) and maintain standard Bellman updates for ID ones ($a \in \mathrm{supp}(\beta)$). That is, the regularization is solely determined by the support of $\beta$, which we refer to as supported value regularization.

However, it is a well-known challenge to distinguish between ID and OOD actions. Directly determining if the behavior density exceeds a threshold [43] would necessitate an extremely precise density estimator, and it is particularly difficult to estimate the density of OOD actions. In this work, we draw inspiration from the link between support and Importance Sampling (IS) to circumvent this dilemma and achieve the same goal.

We begin by explaining its principle with simplified notations. IS computes $l_1 = \mathbb{E}_q[p(x)f(x)/q(x)]$ to estimate $l_2 = \mathbb{E}_p f(x)$. When assuming $\mathrm{supp}(p) \subseteq \mathrm{supp}(q)$, IS is unbiased: $l_1 = l_2$. However, when removing the support assumption, IS actually computes $l_1 = \sum_{x \in \mathrm{supp}(q)} p(x)f(x)$. We observe that $l := l_1 - l_2 = \sum_{x \notin \mathrm{supp}(q)} p(x)f(x)$ gives the summation over the out-of-support region. In the offline RL setting, let $f$ be the $Q$ function. If we choose the behavior policy $\beta$ as $q$ and any samplable distribution that covers the whole action space as $p$, then minimizing $l$ would lower all OOD $Q$-values without affecting the ID ones. Besides, since both distribution $q$ and $p$ are samplable in this case, we can obtain $l_1$ and $l_2$ based on sampling, which provides an unbiased $l$ to optimize.

Therefore, we minimize the following loss for policy evaluation in SVR:

$$\min_{Q} \mathbb{E}_{s\sim\mathcal{D},a\sim\beta} \left[ (Q(s,a) - \mathcal{T}^{\pi}Q'(s,a))^2 \right]$$
$$+ \alpha \left( \mathbb{E}_{s\sim\mathcal{D},a\sim u} \left[ (Q(s,a) - Q_{\min})^2 \right] - \mathbb{E}_{s\sim\mathcal{D},a\sim\beta} \left[ \frac{u(a|s)}{\beta(a|s)}(Q(s,a) - Q_{\min})^2 \right] \right) \quad (5)$$

where $u(\cdot|s)$ is any samplable distribution (e.g., Gaussian, uniform) whose support covers the whole action space, and $Q_{\min} := R_{\min}/(1 - \gamma)$ is the minimal possible $Q$ of the MDP. According to the analysis above, Eq. (5) is equivalent to the following minimization problem:

$$\min_{Q} \mathbb{E}_{s\sim\mathcal{D},a\sim\beta} \left[ (Q(s,a) - \mathcal{T}^{\pi}Q'(s,a))^2 \right] + \alpha\mathbb{E}_{s\sim\mathcal{D}} \left[ \sum_{a\notin\mathrm{supp}(\beta(\cdot|s))} u(a|s)(Q(s,a) - Q_{\min})^2 \right] \quad (6)$$

It is clearer from Eq. (6) that the loss in Eq. (5) penalizes all OOD $Q$-values without affecting the ID ones. Note that optimizing Eq. (5) requires pre-training a behavior model $\beta$ to output the behavior density $\beta(a|s)$ in the IS ratio. Compared with other methods that also require the behavior model [43, 20, 11, 27], SVR is less susceptible to model errors. This is because SVR only needs to query the behavior density of in-dataset $(s,a)$ pairs, thus not requiring much generalization ability of the model, making it relatively easier to estimate accurately.

From an optimization perspective, minimizing Eq. (6) reduces all OOD $Q$-values with a strength proportional to $u(a_{\mathrm{ood}}|s)$. Therefore, we can even choose various $u$ to flexibly penalize the OOD region. For example, let $u$ assign higher weight to areas with a high probability of overestimation. Note that $u$ can have any positive density in the ID region without affecting the ID $Q$-values, thereby eliminating the necessity to manually distinguish between ID and OOD regions. From an optimal solution perspective, Eq. (5,6) lead to the following SVR policy evaluation operator:

$$\mathcal{T}^{\pi}_{\mathrm{SVR}}Q(s,a) = \begin{cases} \mathcal{T}^{\pi}Q(s,a), & \beta(a|s) > 0, \\ Q_{\min}, & \text{else.} \end{cases} \quad (7)$$

In contrast to Observation 1, we make the following claim for SVR.

**Claim 1.** *In each iteration k, compared to $\mathcal{T}^{\pi}$, $\mathcal{T}^{\pi}_{\mathrm{SVR}}$ obtains the same $Q^{k+1}(s,a)$ when $\beta(a|s) > 0$ (ID region); and lowers $Q^{k+1}(s,a)$ when $\beta(a|s) = 0$ (OOD region).*

### 3.3 Analysis

In this section, we refer to the policy iteration, whose evaluation part is repeatedly applying $\mathcal{T}^{\pi}_{\mathrm{SVR}}$ and whose improvement part is vanilla maximization of $Q$, as SVR. We will give a comprehensive analysis of SVR, including the fixed point in policy evaluation, the monotonic improving performance in policy improvement, and the support-constrained optimal convergence for the whole policy iteration.

We first define the support-constrained policy set [22], which plays an important role in our analysis. However, it is worth noting that SVR does not impose any constraint or regularization on the policy.

**Definition 1** (Support-constrained policy)**.** *The support-constrained policy class $\Pi$ is defined as*

$$\Pi = \{\pi \mid \pi(a|s) = 0 \text{ whenever } \beta(a|s) = 0\} \quad (8)$$

Following prior works [22], we also define the optimal support-constrained policy $\pi^*_{\Pi}$.

**Definition 2** (Optimal support-constrained policy)**.** *The optimal support-constrained policy $\pi^*_{\Pi}$ is:*

$$\pi^*_{\Pi}(a|s) := \mathbb{I}\left[ a = \operatorname*{argmax}_{a'\in\mathrm{supp}(\beta(\cdot|s))} Q^*_{\Pi}(s,a') \right] \quad (9)$$

*where $Q^*_{\Pi}$ satisfies the support-constrained Bellman optimality equation:*

$$Q^*_{\Pi}(s,a) = R(s,a) + \gamma\mathbb{E}_{s'\sim P(\cdot|s,a)} \left[ \max_{a'\in\mathrm{supp}(\beta(\cdot|s'))} Q^*_{\Pi}(s',a') \right]. \quad (10)$$

**Proposition 1** (Contraction)**.** *In the whole $\mathcal{S} \times \mathcal{A}$ space and for any $\pi$, $\mathcal{T}^{\pi}_{\mathrm{SVR}}$ is a $\gamma$-contraction operator in the $\mathcal{L}_{\infty}$ norm.*

Therefore, in the policy evaluation stage of SVR, any initial $Q$ function can converge to a unique fixed point by repeatedly applying $\mathcal{T}_{\mathrm{SVR}}^{\pi}$. We give this fixed point in the following theorem.

**Theorem 1** (Fixed point). *SVR yields support-constrained $\pi_i : \pi_i \in \Pi$. The fixed point of $\mathcal{T}_{\mathrm{SVR}}^{\pi_i}$ is*

$$f^{\pi_i}(s,a) = \begin{cases} Q^{\pi_i}(s,a), & \beta(a|s) > 0, \\ Q_{\min}, & \beta(a|s) = 0. \end{cases} \tag{11}$$

*Therefore, for the policies $\pi_i$ in SVR, this fixed point provides unbiased $Q$-values for all ID actions and underestimated $Q$-values for all OOD actions.*

Though Theorem 1 guarantees that SVR only obtains support-constrained policies during learning, we show that, even for any $\pi$ ($\pi \notin \Pi$ due to various errors in practice), the fixed point of $\mathcal{T}_{\mathrm{SVR}}^{\pi}$ still ensures that $Q$ will not be overestimated over the entire action space.

**Proposition 2.** *For any $\pi$, the fixed point of $\mathcal{T}_{\mathrm{SVR}}^{\pi}$ satisfies*

$$\begin{cases} Q_{\min} \le f^{\pi}(s,a) \le Q^{\pi}(s,a), & \beta(a|s) > 0, \\ f^{\pi}(s,a) = Q_{\min}, & \beta(a|s) = 0. \end{cases} \tag{12}$$

For comparison, the following proposition characterizes the contraction property of CQL.

**Proposition 3.** *Only in the ID region ($\beta(a|s) > 0$) and when the evaluated policy is support-constrained ($\pi \in \Pi$), $\mathcal{T}_{\mathrm{CQL}}^{\pi}$ is a contraction operator. Its fixed point satisfies*

$$f^{\pi}(s,a) = Q^{\pi}(s,a) - \frac{\alpha}{1-\gamma}(\rho_{sa}^{\pi})^T\left(\frac{\pi}{\beta} - 1\right), \ \beta(a|s) > 0. \tag{13}$$

Therefore, $\mathcal{T}_{\mathrm{CQL}}^{\pi}$ has no fixed point in the OOD region. Even in the ID region, the condition $\pi \in \Pi$ is required and its fixed point may underestimate or overestimate $Q$-values in a complicated way.

Finally, we show that SVR guarantees strict policy improvement for each iteration until convergence to the optimal support-constrained policy $\pi_\Pi^*$. Note that existing value regularization methods fail to guarantee such each-step policy improvement and optimal convergence results.

**Theorem 2** (Strict policy improvement to support-constrained optimal). *SVR yields support-constrained $\pi_i$ and guarantees monotonic performance improvement:*

$$V^{\pi_{i+1}}(s) \ge V^{\pi_i}(s) \quad \forall s, \tag{14}$$

*where the improvement is strict in at least one state until $\pi_\Pi^*$ is found.*

### 3.4 Practical implementation of SVR

SVR is easy to implement and we design the practical algorithm to be as simple as possible to avoid some complex modules confusing our algorithm's impact on the final performance [1].

**Behavior model.** Following previous works [22, 44], we learn a Gaussian model $\beta_\omega$ for the behavior policy by maximizing

$$J_\beta(\omega) = \mathbb{E}_{s,a\sim\mathcal{D}} \log \beta_\omega(a|s) \tag{15}$$

**Sampling distribution $u$.** In our method, $u$ in Eq. (5) can be any samplable distribution that covers the entire action space. In our implementation, we set $u$ to the Gaussian with the same mean as $\pi$ and a fixed variance that is much larger than that of $\pi$. This choice is made because during learning, $\pi$ is typically

---

**Algorithm 1** Supported Value Regularization (SVR)

1: Initialize behavior policy $\beta_\omega$, policy network $\pi_\phi$, $Q$-network $Q_\theta$, and target $Q$-network $Q_{\theta'}$
2: **// Behavior Policy Pre-training**
3: **for** each gradient step **do**
4:      Sample minibatch $(s,a) \sim \mathcal{D}$
5:      Update $\omega$ by maximizing $J_\beta(\omega)$ in Eq. (15)
6: **end for**
7: **// Policy Training**
8: **for** each gradient step **do**
9:      Sample minibatch $(s,a,r,s') \sim \mathcal{D}$
10:      Update $\theta$ by minimizing $L_Q(\theta)$ in Eq. (16)
11:      Update $\phi$ by minimizing $L_\pi(\phi)$ in Eq. (2)
12:      Update target network: $\theta' \leftarrow (1-\tau)\theta' + \tau\theta$
13: **end for**

---

located where $Q$ is maximized and where overestimation is most likely to occur. So if $\pi$ is OOD, this choice will assign higher weight to the area around $\pi$ to reduce the $Q$-value. Note that if $\pi$ is ID, it will not penalize the $Q$-values in the ID region even though $u$ may have a higher weight there.

---

[1] Our code is available at https://github.com/MAOYIXIU/SVR.

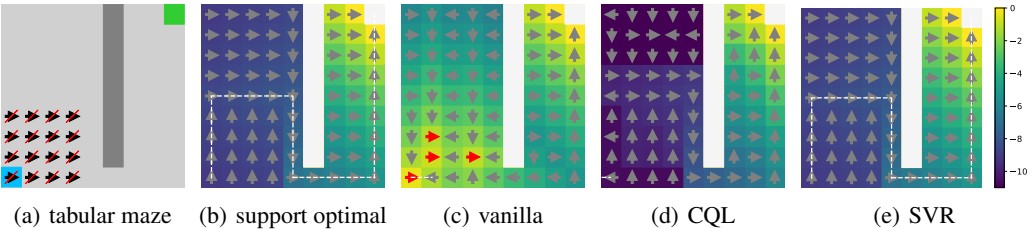

|  (a) tabular maze | (b) support optimal | (c) vanilla | (d) CQL | (e) SVR |

Figure 1: (a) The maze environment with OOD actions. (b) The optimal support-constrained policy and its value function. (c) (d) (e) The value functions and policies learned by vanilla policy iteration, CQL and SVR, respectively. The resulting trajectories are illustrated by white dashed lines. SVR is capable of obtaining the optimal support-constrained policy and value function.

**Policy evaluation.** With policy $\pi_\phi$, $Q$ function $Q_\theta$, and target $Q$ function $Q_{\theta'}$, we minimize the following loss for policy evaluation:

$$
\begin{aligned}
L_Q(\theta) = & \mathbb{E}_{(s,a,s')\sim\mathcal{D}}[(Q_\theta(s,a) - R(s,a) - \gamma\mathbb{E}_{a'\sim\pi_\phi(\cdot|s')}Q_{\theta'}(s',a'))^2] \\
& + \alpha\left(\mathbb{E}_{s\sim\mathcal{D},a\sim u}\left[(Q_\theta(s,a) - Q_{\min})^2\right] - \mathbb{E}_{s,a\sim\mathcal{D}}\left[\frac{u(a|s)}{\beta_\omega(a|s)}(Q_\theta(s,a) - Q_{\min})^2\right]\right)
\end{aligned}
\tag{16}
$$

where $\alpha$ is a hyperparameter and $Q_{\min} := R_{\min}/(1 - \gamma)$. For the environment where $R_{\min}$ is unknown, we choose the smallest reward in the dataset as $R_{\min}$.

**Overall Algorithm.** Putting everything together, we summarize our final algorithm in Algorithm 1. Our algorithm first trains the estimated behavior policy to obtain the behavior density. Then it turns to the actor-critic framework for policy training.

## 4 Experiments

In this section, we conduct several experiments to justify the validity of our method. We aim to answer five questions: (1) Does SVR actually converge to the optimal support-constrained policy? (2) Does SVR perform better than previous methods on standard offline RL benchmarks? (3) When does SVR empirically benefit the most compared to the density-based regularization? (4) How should we select the sampling distribution of SVR in practice? (5) How does the implementation of each component affect SVR? More experimental details and results are provided in Appendix B and C.

### 4.1 Support-constrained optimality in the tabular setting

We use a simple maze environment to verify the support-constrained optimality of SVR. As depicted in Fig. 1(a), the task is to navigate from bottom-left to top-right, with a wall in the middle. The agent receives a reward of 0 for reaching the goal and $-1$ for all other transitions. Episodes are terminated after 100 steps and $\gamma$ is set to 0.9. We first collect $10,000$ transitions using a random policy. Then we remove all the transitions containing rightward actions in the bottom-left $4 \times 4$ region to introduce OOD actions. It makes the optimal support-constrained policy (see Fig. 1(b)) differ from the actual optimal policy in the environment.

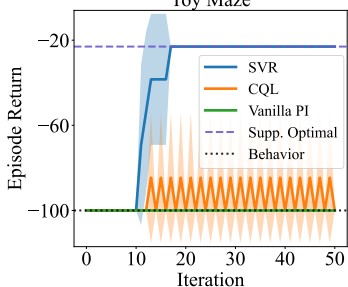

Figure 2: Learning curves of SVR, CQL and vanilla policy iteration on the toy maze.

We test three algorithms: vanilla policy iteration [41], CQL [23], and SVR. Fig. 1(c, d, e) depict their learned value functions and policies respectively. To show convergence results, we also present the learning curves in Fig. 2, along with the performance of the behavior policy and the optimal support-constrained policy. The results indicate that vanilla policy iteration has a severe overestimation of value functions, leading to poor performance. On the other hand, although CQL does not overestimate $V$ functions, it fails to converge and performs poorly when the dataset is highly suboptimal. In contrast, SVR converges to the optimal support-constrained policy and the learned value function closely matches the true support-constrained optimal value function, verifying Theorem 2.

Table 1: Averaged normalized scores on the D4RL benchmarks over five random seeds. Note that m = medium, m-r = medium-replay, m-e = medium-expert, e = expert, r = random.

| Dataset | BC | OneStep | TD3BC | AWAC | BCQ | BEAR | UWAC | CQL | IQL | SVR (Ours) |
|---|---|---|---|---|---|---|---|---|---|---|
| halfcheetah-m | 42.0 | 50.4 | 48.3 | 47.9 | 46.6 | 43.0 | 42.2 | 47.0 | 47.4 | **60.5±1.2** |
| hopper-m | 56.2 | 87.5 | 59.3 | 59.8 | 59.4 | 51.8 | 50.9 | 53.0 | 66.2 | **103.5±0.4** |
| walker2d-m | 71.0 | 84.8 | 83.7 | 83.1 | 71.8 | -0.2 | 75.4 | 73.3 | 78.3 | **92.4±1.2** |
| halfcheetah-m-r | 36.4 | 42.7 | 44.6 | 44.8 | 42.2 | 36.3 | 35.9 | 45.5 | 44.2 | **52.5±3.0** |
| hopper-m-r | 21.8 | 98.5 | 60.9 | 69.8 | 60.9 | 52.2 | 25.3 | 88.7 | 94.7 | **103.7±1.3** |
| walker2d-m-r | 24.9 | 61.7 | 81.8 | 78.1 | 57.0 | 7.0 | 23.6 | 81.8 | 73.8 | **95.6±2.5** |
| halfcheetah-m-e | 59.6 | 75.1 | 90.7 | 64.9 | **95.4** | 46.0 | 42.7 | 75.6 | 86.7 | 94.2±2.2 |
| hopper-m-e | 51.7 | 108.6 | 98.0 | 100.1 | 106.9 | 50.6 | 44.9 | 105.6 | 91.5 | **111.2±0.9** |
| walker2d-m-e | 101.2 | **111.3** | 110.1 | 110.0 | 107.7 | 22.1 | 96.5 | 107.9 | 109.6 | 109.3±0.2 |
| halfcheetah-e | 92.9 | 88.2 | **96.7** | 81.7 | 89.9 | 92.7 | 92.9 | 96.3 | 95.0 | 96.1±0.7 |
| hopper-e | **110.9** | 106.9 | 107.8 | 109.5 | 109.0 | 54.6 | 110.5 | 96.5 | 109.4 | 111.1±0.4 |
| walker2d-e | 107.7 | **110.7** | 110.2 | 110.1 | 106.3 | 106.6 | 108.4 | 108.5 | 109.9 | 110.0±0.2 |
| halfcheetah-r | 2.6 | 2.3 | 11.0 | 6.1 | 2.2 | 2.3 | 2.3 | 17.5 | 13.1 | **27.2±1.2** |
| hopper-r | 4.1 | 5.6 | 8.5 | 9.2 | 7.8 | 3.9 | 2.7 | 7.9 | 7.9 | **31.0±0.3** |
| walker2d-r | 1.2 | 6.9 | 1.6 | 0.2 | 4.9 | **12.8** | 2.0 | 5.1 | 5.4 | 2.2±1.5 |
| gym-v2 total | 784.2 | 1041.2 | 1013.2 | 975.6 | 968.0 | 581.7 | 756.2 | 1010.2 | 1033.1 | **1200.5** |
| pen-expert | 85.1 | 61.6 | 111.0 | 115.2 | 114.9 | 105.9 | 111.9 | 107.0 | 110.2 | **138.9±9.2** |
| pen-human | 34.4 | **73.7** | 54.9 | 25.5 | 68.9 | -1.0 | 21.7 | 37.5 | 71.5 | 73.1±12.1 |
| pen-cloned | 56.9 | 31.8 | 63.8 | 10.4 | 44.0 | 26.5 | 33.1 | 39.2 | 37.3 | **70.2±17.4** |
| adroit-v0 total | 176.4 | 167.1 | 229.7 | 151.1 | 227.8 | 131.4 | 166.7 | 183.7 | 219.0 | **282.2** |

## 4.2 Comparisons on D4RL benchmarks

Then we evaluate our approach on the D4RL benchmarks [7]. We compare SVR with prior state-of-the-art offline RL methods, including BC [37], BCQ [10], BEAR [22], OneStep RL [4], TD3+BC [8], AWAC [33], UWAC [45], CQL [23], and IQL [21].

The results are reported in Table 1. In both Gym-MuJoCo and Adroit domains, SVR achieves state-of-the-art performance and outperforms prior value regularization methods (CQL, UWAC) by a large margin. We also observe that SVR has strong advantages on the sub-optimal datasets (medium, medium-replay, random). This is because SVR exploits the optimal support-constrained policy in the dataset in a theoretically sound way and is less affected by the average quality of the dataset. In addition, while BCQ and BEAR are the policy constraint methods that are also designed to search for the optimal policy within the behavior support, they have much poorer performance. As stated in previous works [11, 44], they are limited respectively by the errors of the generative model and the unsuitability of using Maximum Mean Discrepancy (MMD) to characterize the support constraint. In contrast, without querying out-of-dataset behavior density, SVR takes the form of value regularization to search for the optimal support-constrained policy and achieves superior performance. For learning curves and more experimental details, please see Section B in the supplementary material.

## 4.3 Comparisons on noisy datasets.

In this section, we aim to validate that, compared with the existing density-based value regularization, the supported value regularization of SVR will benefit when the dataset contains a large portion of bad actions. To this end, we construct a "noisy" dataset by combining the random dataset and the expert dataset, and then evaluate SVR and CQL under different expert ratios.

The results are shown in Fig. 3. In all environments, SVR outperforms CQL over nearly all expert ratios. Moreover, as the expert ratio gets lower, the improvement is more significant. The performance of CQL is susceptible to the expert ratio and exhibits a sharp decrease at the expert ratio 20%, while SVR can still retain good or even expert performance (most spark in Hopper and Walker2d).

## 4.4 Empirical study on the sampling distribution of SVR

In this section, we investigate how to practically select the sampling distribution $u$ in SVR, which controls the relative strength of penalizing different OOD $Q$-values. In our implementation, $u$ is the

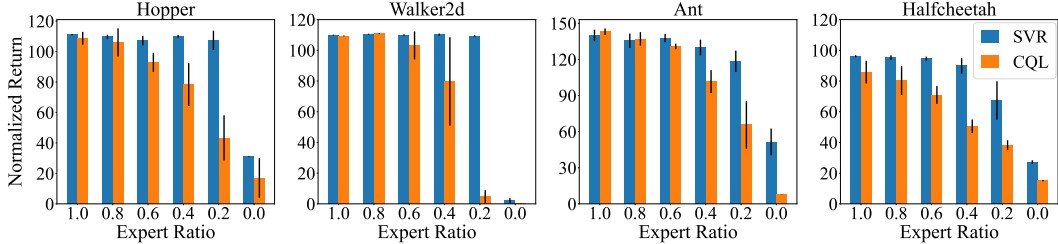

Figure 3: Evaluations of SVR (supported value regularization) vs CQL (density value regularization) on noisy datasets, which are made by mixing random and expert datasets with varying expert ratios.

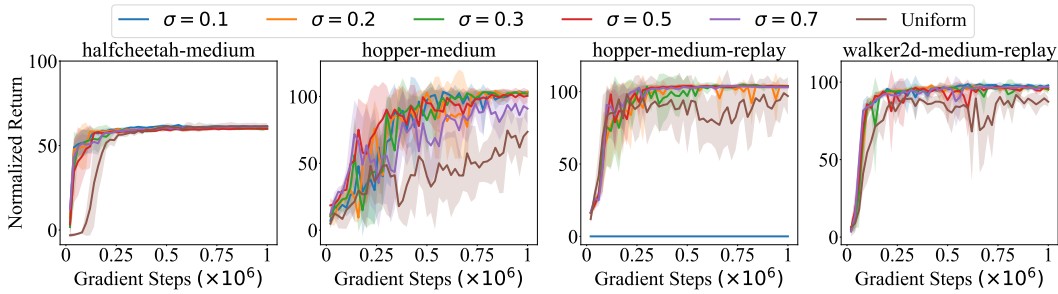

Figure 4: Learning curves of SVR with different sampling distribution $u$. $\sigma$ is the standard deviation of Gaussian. SVR is able to converge to good performance under various sampling distributions.

Gaussian with the same mean as the current policy. Here we vary its standard deviation $\sigma$ and present the corresponding results in Fig. 4. We do not show the results for $\sigma > 0.7$ because such a large $\sigma$ would cause most sampled actions to lie at the boundary of the action space. Instead, we use Uniform distribution to represent $\sigma = \infty$.

As shown in Fig. 4, SVR is able to converge to good performance over a very wide range of $\sigma$. However, if $\sigma$ is too small, SVR may not be able to adequately penalize all OOD $Q$-values and the IS ratio will have a large variance, thereby disrupting the learning process (see $\sigma = 0.1$ in hopper-m-r). Conversely, if $\sigma$ is too large or if the uniform distribution is used, sampling from $u$ may fail to emphasize the key areas where overestimation is most likely to occur (as indicated by the current policy), leading to insufficient mitigation of overestimation and inferior performance (see Uniform in hopper-m, hopper-m-r, walker2d-m-r and $\sigma = 0.7$ in hopper-m).

### 4.5 Effects of components in SVR

**Density estimator.** We speculate that a more precise behavior density estimator may further improve SVR. So following previous works [10, 43], we consider to replace the Gaussian density estimator $\beta_\omega$ with the conditional variational autoencoder [18, 40]. We refer to this variant as SVR-VAE. The results of SVR-VAE are shown in Fig. 5 (left). Overall, SVR-VAE has only marginal performance gains over SVR.

**IS techniques.** Self-Normalized Importance Sampling (SNIS) [35] is sometimes adopted to reduce the high variance

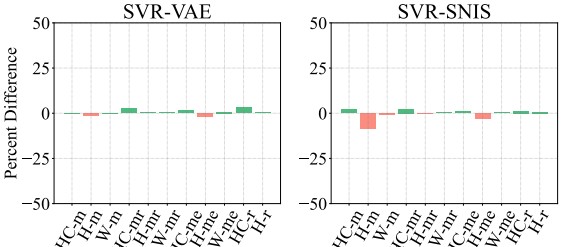

Figure 5: Percentage performance difference of the SVR variant compared to the original algorithm. (Left) SVR with VAE density estimator. (Right) SVR with self-normalized importance sampling. HC = HalfCheetah, H = Hopper, W = Walker2d.

of IS [38]. The SNIS estimator has lower variance but is biased [35]. Here we test an SVR variant SVR-SNIS, which normalizes the IS ratio across the batch. As shown in Fig. 5 (right), the perfor-

mance of SVR-SNIS is worse in hopper-m and hopper-m-e, and slightly better in other tasks. This discrepancy may be due to the varying requirements for variance and bias across different tasks.

## 5 Related Work

**Offline RL.** In offline RL, extrapolation error and overestimation caused by OOD actions pose significant challenges [10]. Among various solutions, value regularization methods aim to introduce conservatism in value estimation [23, 20, 29, 6, 3, 27], while policy constraint approaches enforce proximity between the trained policy and the behavior policy, either explicitly via divergence penalties [44, 22, 13, 8], implicitly by weighted behavior cloning [5, 36, 33, 42, 31], or directly through specific parameterization of the policy [10, 11, 53]. Another line of the methods, instead, opt for in-sample learning, which involves formulating the Bellman target with only the actions in the dataset [4, 28, 21, 51, 47]. However, the performance of most existing methods is largely confined by the average quality of the dataset [25]. In contrast, SVR guarantees the convergence to the optimal support-constrained policy and achieve superior performance in sub-optimal datasets.

**Value regularization.** We divide the value regularization methods in offline RL into two categories: direct $Q$ penalization and Bellman target reduction. The former involves adding penalties to the critic learning loss, trying to penalize OOD $Q$-values [23, 50, 46, 6, 27]. Due to the difficulty of determining the OOD region, they typically penalize $Q$-values under the current policy distribution [23, 3, 27, 48] and optionally incorporate a maximization term under the data distribution for milder pessimism [23, 50, 6]. Thus, existing works of this category belong to policy density based value regularization, rather than ID/OOD based. On the other hand, the Bellman target reduction methods either subtract an uncertainty quantifier from the Bellman target [14, 3] or directly use the minimum of $Q$ ensembles [34] to compute the target [2, 9]. However, if not incorporating additional penalties, they do not provide learning signals on OOD $Q$-values [2]. It empirically works because after the minimization over ensemble (or subtracting the uncertainty quantifier), the random generalized OOD $Q$-values will be smaller than ID ones with high probability, reducing the impact of OOD $Q$-values on learning. However, it requires a large number of $Q$ networks and is much more computationally expensive [2].

**Support-constrained optimality.** In absence of additional information beyond the dataset, the optimal support-constrained policy represents the best policy that we can hope to obtain. Recent works have considered various policy constraints to learn it, but their performance still leaves considerable room for improvement. For example, BEAR [22] attempts to keep the learned policy within the support of the behavior policy by minimizing MMD between them. However, this choice lacks a theoretical guarantee, and Wu et al. [44] empirically found that MMD has no performance gain over KL. Other works adopt a specific parameterization of the policy to constrain it to the behavior support [10, 11]. They first pre-train a generative model (e.g., VAE) for the behavior policy. Then during learning, they sample several actions at each state and choose the one with the highest $Q$-value as the output of the policy, aiming to obtain the maximum within the behavior support. However, these methods are vulnerable to model errors [11] and have a high computational cost for generating sufficient actions for each state. Instead of working on policy constraints, SVR is the first value regularization algorithm to achieve support-constrained optimality. Moreover, SVR only needs to query the behavior density of in-dataset $(s, a)$ pairs, making it less susceptible to model errors.

## 6 Conclusions and Limitations

In this work, we propose a novel value regularization method, SVR, which penalizes the $Q$-values for all OOD actions while maintaining standard Bellman updates for ID ones. We show that the policy evaluation operator of SVR is a contraction, whose fixed point outputs unbiased $Q$-values in ID region and underestimated $Q$-values in OOD region. Furthermore, SVR guarantees strict policy improvement until convergence to the optimal support-constrained policy. Empirical results validate the theoretical properties of SVR and demonstrate its SoTA performance on the D4RL benchmarks.

One limitation of SVR lies in the need of pre-training the behavior model. An exciting direction for future work would be to achieve supported value regularization without explicit behavior policy estimation.

## Acknowledgment

This work was supported by the National Key R&D Program of China under Grant 2018AAA0102801, National Natural Science Foundation of China under Grant 61827804.

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

# A Proofs

In this section, we present the proofs for all the theories in the paper.

We first restate the two definitions in the paper.

**Definition 3** (Support-constrained policy, Definition 1). *The support-constrained policy class $\Pi$ is defined as*

$$\Pi = \{\pi \mid \pi(a|s) = 0 \text{ whenever } \beta(a|s) = 0\} \tag{17}$$

**Definition 4** (Optimal support-constrained policy, Definition 2). *The optimal support-constrained policy $\pi_\Pi^*$ is:*

$$\pi_\Pi^*(a|s) := \mathbb{I}\left[a = \operatorname*{argmax}_{a' \in \operatorname{supp}(\beta(\cdot|s))} Q_\Pi^*(s, a')\right] \tag{18}$$

*where $Q_\Pi^*$ satisfies the support-constrained Bellman optimality equation:*

$$Q_\Pi^*(s, a) = R(s, a) + \gamma \mathbb{E}_{s' \sim P(\cdot|s,a)}\left[\max_{a' \in \operatorname{supp}(\beta(\cdot|s'))} Q_\Pi^*(s', a')\right]. \tag{19}$$

**Proposition 4** (Contraction, Proposition 1). *In the whole $\mathcal{S} \times \mathcal{A}$ space and for any $\pi$, $\mathcal{T}_{\mathrm{SVR}}^\pi$ is a $\gamma$-contraction operator in the $\mathcal{L}_\infty$ norm.*

*Proof.*

$$\mathcal{T}_{\mathrm{SVR}}^\pi Q(s, a) = \begin{cases} \mathcal{T}^\pi Q(s, a), & \beta(a|s) > 0, \\ Q_{\min}, & \text{else.} \end{cases} \tag{20}$$

Let $f_1$ and $f_2$ be two arbitrary functions.

For all $(s, a)$ s.t. $\beta(a|s) = 0$, we have

$$|\mathcal{T}_{\mathrm{SVR}}^\pi f_1(s, a) - \mathcal{T}_{\mathrm{SVR}}^\pi f_2(s, a)| = Q_{\min} - Q_{\min} \le \gamma \|f_1 - f_2\|_\infty \tag{21}$$

For all $(s, a)$ s.t. $\beta(a|s) > 0$, we have

$$
\begin{aligned}
|\mathcal{T}_{\mathrm{SVR}}^\pi f_1(s, a) - \mathcal{T}_{\mathrm{SVR}}^\pi f_2(s, a)| &= \left|\gamma \mathbb{E}_{s' \sim P(\cdot|s,a), a' \sim \pi(\cdot|s)}[f_1(s', a') - f_2(s', a')]\right| \\
&\le \gamma \mathbb{E}_{s' \sim P(\cdot|s,a), a' \sim \pi(\cdot|s)}[|f_1(s', a') - f_2(s', a')|] \\
&\le \gamma \|f_1 - f_2\|_\infty
\end{aligned} \tag{22}
$$

Therefore, we have $\|\mathcal{T}_{\mathrm{SVR}}^\pi f_1 - \mathcal{T}_{\mathrm{SVR}}^\pi f_2\|_\infty \le \gamma \|f_1 - f_2\|_\infty$. $\qquad\square$

**Proposition 5** (Proposition 2). *For any $\pi$, the fixed point of $\mathcal{T}_{\mathrm{SVR}}^\pi$ satisfies*

$$\begin{cases} Q_{\min} \le f^\pi(s, a) \le Q^\pi(s, a), & \beta(a|s) > 0, \\ f^\pi(s, a) = Q_{\min}, & \beta(a|s) = 0. \end{cases} \tag{23}$$

*Proof.* By Proposition 4, $\mathcal{T}_{\mathrm{SVR}}^\pi$ is a contraction and has a unique fixed point. Assume the fixed point is $f^\pi$. Thus,

$$f^\pi(s, a) = \mathcal{T}_{\mathrm{SVR}}^\pi f^\pi(s, a) = \begin{cases} \mathcal{T}^\pi f^\pi(s, a), & \beta(a|s) > 0, \\ Q_{\min}, & \text{else.} \end{cases} \tag{24}$$

Define $(\hat{s}, \hat{a}) := \operatorname{argmin}_{s \in \mathcal{S}, a \in \operatorname{supp}(\beta(\cdot|s))} f^\pi(s, a)$, and $p^\pi(s, a) := \mathbb{E}_{s' \sim P(\cdot|s,a), a' \sim \pi(\cdot|s')} \pi(a'|s')$.

As $\beta(\hat{a}|\hat{s}) > 0$,

$$
\begin{aligned}
f^\pi(\hat{s}, \hat{a}) &= \mathcal{T}^\pi_{\mathrm{SVR}} f^\pi(\hat{s}, \hat{a}) \\
&= r(\hat{s}, \hat{a}) + \gamma \mathbb{E}_{\hat{s}'} \mathbb{E}_{\hat{a}' \sim \pi(\cdot|\hat{s}')} [f^\pi(\hat{s}', \hat{a}')] \\
&= r(\hat{s}, \hat{a}) + \gamma \mathbb{E}_{\hat{s}'} \mathbb{E}_{\hat{a}' \sim \pi(\cdot|\hat{s}')} [\mathcal{T}^\pi_{\mathrm{SVR}} f^\pi(\hat{s}', \hat{a}')] \\
&= r(\hat{s}, \hat{a}) + \gamma \mathbb{E}_{\hat{s}'} \left[ \sum_{\hat{a}' \in \mathrm{supp}(\beta(\cdot|\hat{s}'))} \pi(\hat{a}'|\hat{s}') \mathcal{T}^\pi f^\pi(\hat{s}', \hat{a}') + \sum_{\hat{a}' \notin \mathrm{supp}(\beta(\cdot|\hat{s}'))} \pi(\hat{a}'|\hat{s}') Q_{\min} \right] \\
&= r(\hat{s}, \hat{a}) + \gamma \mathbb{E}_{\hat{s}'} \left[ \sum_{\hat{a}' \in \mathrm{supp}(\beta(\cdot|\hat{s}'))} \pi(\hat{a}'|\hat{s}') f^\pi(\hat{s}', \hat{a}') + \sum_{\hat{a}' \notin \mathrm{supp}(\beta(\cdot|\hat{s}'))} \pi(\hat{a}'|\hat{s}') Q_{\min} \right] \\
&\geq r_{\min} + \gamma \mathbb{E}_{\hat{s}'} \left[ \sum_{\hat{a}' \in \mathrm{supp}(\beta(\cdot|\hat{s}'))} \pi(\hat{a}'|\hat{s}') f^\pi(\hat{s}, \hat{a}) + \sum_{\hat{a}' \notin \mathrm{supp}(\beta(\cdot|\hat{s}'))} \pi(\hat{a}'|\hat{s}') Q_{\min} \right]
\end{aligned}
$$

The last inequality holds because $f^\pi(\hat{s}, \hat{a}) := \min_{s \in \mathcal{S}, a \in \mathrm{supp}(\beta(\cdot|s))} f^\pi(s, a)$.

Now we define a shorthand

$$
\lambda := \mathbb{E}_{\hat{s}'} \left[ \sum_{\hat{a}' \in \mathrm{supp}(\beta(\cdot|\hat{s}'))} \pi(\hat{a}'|\hat{s}') \right]
$$

It follows that

$$
f^\pi(\hat{s}, \hat{a}) \geq r_{\min} + \gamma \lambda f^\pi(\hat{s}, \hat{a}) + \gamma(1 - \lambda) Q_{\min}
$$

Because $\lambda \in [0, 1]$, we have

$$
\begin{aligned}
f^\pi(\hat{s}, \hat{a}) &\geq \frac{r_{\min} + \gamma(1 - \lambda) Q_{\min}}{1 - \gamma \lambda} \\
&= \frac{(1 - \gamma) Q_{\min} + \gamma(1 - \lambda) Q_{\min}}{1 - \gamma \lambda} \\
&= Q_{\min}
\end{aligned}
$$

Therefore, for all $(s, a)$ s.t. $\beta(a|s) > 0$, it holds that $f^\pi(s, a) \geq f^\pi(\hat{s}, \hat{a}) \geq Q_{\min}$. Besides, For $(s, a)$ s.t. $\beta(a|s) = 0$, it holds that $f^\pi(s, a) = Q_{\min}$. Thus $f^\pi(s, a) \geq Q_{\min}, \forall s, a$.

Before we prove $f^\pi(s, a) \leq Q^\pi(s, a)$ when $\beta(a|s) > 0$, we first prove $\mathcal{T}^\pi_{\mathrm{SVR}} f^\pi(s, a) \leq \mathcal{T}^\pi f^\pi(s, a), \forall s, a$

For any $(s, a)$, We have

$$
\begin{aligned}
\mathcal{T}^\pi f^\pi(s, a) &= r(s, a) + \gamma \mathbb{E}_{s'} \mathbb{E}_{a' \sim \pi(\cdot|s')} [f^\pi(s', a')] \\
&\geq r_{\min} + \gamma \mathbb{E}_{s'} \mathbb{E}_{a' \sim \pi(\cdot|s')} [Q_{\min}] \\
&= Q_{\min}
\end{aligned}
\tag{25}
$$

Therefore,

$$
\mathcal{T}^\pi_{\mathrm{SVR}} f^\pi(s, a) = \begin{cases} \mathcal{T}^\pi f^\pi(s, a) \leq \mathcal{T}^\pi f^\pi(s, a), & \beta(a|s) > 0, \\ Q_{\min} \leq \mathcal{T}^\pi f^\pi(s, a), & \text{else.} \end{cases}
\tag{26}
$$

Thus, it holds that $\mathcal{T}^\pi_{\mathrm{SVR}} f^\pi(s, a) \leq \mathcal{T}^\pi f^\pi(s, a), \forall s, a$.

Now we prove $f^\pi(s,a) \le Q^\pi(s,a)$ when $\beta(a|s) > 0$. For $\beta(a|s) > 0$,

$$
\begin{aligned}
f^\pi(s,a) &= \mathcal{T}_{\mathrm{SVR}}^\pi f^\pi(s,a) \\
&= r(s,a) + \gamma \mathbb{E}_{s'} \mathbb{E}_{a' \sim \pi(\cdot|s')}[f^\pi(s',a')] \\
&= r(s,a) + \gamma \mathbb{E}_{s'} \mathbb{E}_{a' \sim \pi(\cdot|s')}[\mathcal{T}_{\mathrm{SVR}}^\pi f^\pi(s',a')] \\
&\le r(s,a) + \gamma \mathbb{E}_{s'} \mathbb{E}_{a' \sim \pi(\cdot|s')}[\mathcal{T}^\pi f^\pi(s',a')] \\
&= r(s,a) + \gamma \mathbb{E}_{s'} \mathbb{E}_{a' \sim \pi(\cdot|s')}[r(s',a') + \gamma \mathbb{E}_{s''} \mathbb{E}_{a'' \sim \pi(\cdot|s'')}[f^\pi(s'',a'')]] \\
&= r(s,a) + \gamma \mathbb{E}_{s'} \mathbb{E}_{a' \sim \pi(\cdot|s')}[r(s',a') + \gamma \mathbb{E}_{s''} \mathbb{E}_{a'' \sim \pi(\cdot|s'')}[\mathcal{T}_{\mathrm{SVR}}^\pi f^\pi(s'',a'')]] \\
&\le r(s,a) + \gamma \mathbb{E}_{s'} \mathbb{E}_{a' \sim \pi(\cdot|s')}[r(s',a') + \gamma \mathbb{E}_{s''} \mathbb{E}_{a'' \sim \pi(\cdot|s'')}[\mathcal{T}^\pi f^\pi(s'',a'')]] \\
&\cdots \\
&\le \mathbb{E}_\pi \left[ \sum_{j=0}^\infty \gamma^j r(s_{t+j}, a_{t+j}) | s_t = s, a_t = a \right] \\
&= Q^\pi(s,a)
\end{aligned}
$$

In conclusion, for any $\pi$, the fixed point of $\mathcal{T}_{\mathrm{SVR}}^\pi$ satisfies

$$
\begin{cases}
Q_{\min} \le f^\pi(s,a) \le Q^\pi(s,a), & \beta(a|s) > 0, \\
f^\pi(s,a) = Q_{\min}, & \beta(a|s) = 0.
\end{cases}
\tag{27}
$$

□

**Theorem 3** (Fixed point, Theorem 1). *SVR yields support-constrained $\pi_i : \pi_i \in \Pi$. The fixed point of $\mathcal{T}_{\mathrm{SVR}}^{\pi_i}$ is*

$$
f^{\pi_i}(s,a) = \begin{cases}
Q^{\pi_i}(s,a), & \beta(a|s) > 0, \\
Q_{\min}, & \beta(a|s) = 0.
\end{cases}
\tag{28}
$$

*Therefore, for the policies $\pi_i$ in SVR, this fixed point provides unbiased Q-values for all ID actions and underestimated Q-values for all OOD actions.*

*Proof.* By Proposition 4, $\mathcal{T}_{\mathrm{SVR}}^\pi$ is a contraction and has a unique fixed point. For any $\pi_i$, the fixed point of policy evaluation satisfies $f^{\pi_i} = \mathcal{T}_{\mathrm{SVR}}^\pi f^{\pi_i}$.

According to Proposition 5, the policy improvement at this iteration satisfies

$$
\pi_{i+1}(a|s) = \mathbb{I}\left[ a = \operatorname*{argmax}_{a'} f^{\pi_i}(s,a') \right] = \mathbb{I}\left[ a = \operatorname*{argmax}_{a' \in \mathrm{supp}(\beta(\cdot|s))} f^{\pi_i}(s,a') \right]
\tag{29}
$$

Therefore, for arbitrary initial policy $\pi_0$, $\pi_i$ $(i > 0)$ in SVR is support-constrained: $\pi_i \in \Pi, i > 0$.

Now we show the fixed point of $\mathcal{T}_{\mathrm{SVR}}^{\pi_i}$ $(i > 0)$. Assume the fixed point is $f^{\pi_i}$:

$$
f^{\pi_i}(s,a) = \mathcal{T}_{\mathrm{SVR}}^{\pi_i} f^{\pi_i}(s,a) = \begin{cases}
\mathcal{T}^{\pi_i} f^{\pi_i}(s,a), & \beta(a|s) > 0, \\
Q_{\min}, & \text{else.}
\end{cases}
\tag{30}
$$

For $\beta(a|s) > 0$,

$$
\begin{aligned}
f^{\pi_i}(s,a) &= \mathcal{T}_{\mathrm{SVR}}^{\pi_i} f^{\pi_i}(s,a) \\
&= r(s,a) + \gamma \mathbb{E}_{s'} \mathbb{E}_{a' \sim \pi_i(\cdot|s')}[f^{\pi_i}(s',a')] \\
&= r(s,a) + \gamma \mathbb{E}_{s'} \mathbb{E}_{a' \sim \pi_i(\cdot|s')}[\mathcal{T}_{\mathrm{SVR}}^{\pi_i} f^{\pi_i}(s',a')] \\
&= r(s,a) + \gamma \mathbb{E}_{s'} \mathbb{E}_{a' \sim \pi_i(\cdot|s')}[r(s',a') + \gamma \mathbb{E}_{s''} \mathbb{E}_{a'' \sim \pi_i(\cdot|s'')}[f^{\pi_i}(s'',a'')]] \\
&= r(s,a) + \gamma \mathbb{E}_{s'} \mathbb{E}_{a' \sim \pi_i(\cdot|s')}[r(s',a') + \gamma \mathbb{E}_{s''} \mathbb{E}_{a'' \sim \pi_i(\cdot|s'')}[\mathcal{T}_{\mathrm{SVR}}^{\pi_i} f^{\pi_i}(s'',a'')]] \\
&\cdots \\
&= \mathbb{E}_{\pi_i} \left[ \sum_{j=0}^\infty \gamma^j r(s_{t+j}, a_{t+j}) | s_t = s, a_t = a \right] \\
&= Q^{\pi_i}(s,a)
\end{aligned}
$$

The fourth equality holds because $\pi_i$ is support-constrained and thus the expectation $\mathbb{E}_{a' \sim \pi_i(\cdot|s')}$ has non-zero mass only on support-constrained $a'$, i.e., $\beta(a'|s') > 0$.

Therefore, for the policies $\pi_i$ ($i > 0$) in SVR, the fixed point of $\mathcal{T}_{\mathrm{SVR}}^{\pi_i}$ is

$$f^{\pi_i}(s,a) = \begin{cases} Q^{\pi_i}(s,a), & \beta(a|s) > 0, \\ Q_{\min}, & \beta(a|s) = 0. \end{cases} \tag{31}$$

This fixed point provides unbiased $Q$-values for all ID actions and underestimated $Q$-values for all OOD actions. $\qquad \square$

**Proposition 6** (Proposition 3)**.** *Only in the ID region ($\beta(a|s) > 0$) and when the evaluated policy is support-constrained ($\pi \in \Pi$), $\mathcal{T}_{\mathrm{CQL}}^{\pi}$ is a contraction operator. Its fixed point satisfies*

$$f^{\pi}(s,a) = Q^{\pi}(s,a) - \frac{\alpha}{1-\gamma}(\rho_{sa}^{\pi})^T \left( \frac{\pi}{\beta} - 1 \right), \; \beta(a|s) > 0. \tag{32}$$

*Proof.*

$$\mathcal{T}_{\mathrm{CQL}}^{\pi} Q(s,a) = \begin{cases} \mathcal{T}^{\pi} Q(s,a) - \alpha \left( \frac{\pi(a|s)}{\beta(a|s)} - 1 \right), & \beta(a|s) > 0, \\ -\infty, & \beta(a|s) = 0, \pi(a|s) > 0, \\ Q(s,a), & \beta(a|s) = 0, \pi(a|s) = 0. \end{cases} \tag{33}$$

Let $f_1$ and $f_2$ be two arbitrary functions.

We first consider the OOD region ($\beta(a|s) = 0$). We separate it into $\pi(a|s) = 0$ and $\pi(a|s) > 0$.

For all $(s,a)$ s.t. $\beta(a|s) = 0$ and $\pi(a|s) = 0$,

$$|\mathcal{T}_{\mathrm{CQL}}^{\pi} f_1(s,a) - \mathcal{T}_{\mathrm{CQL}}^{\pi} f_2(s,a)| = |f_1(s,a) - f_2(s,a)| \tag{34}$$

For all $(s,a)$ s.t. $\beta(a|s) = 0$ and $\pi(a|s) > 0$,

$$|\mathcal{T}_{\mathrm{CQL}}^{\pi} f_1(s,a) - \mathcal{T}_{\mathrm{CQL}}^{\pi} f_2(s,a)| = |\infty - \infty| \tag{35}$$

In both cases, $\mathcal{T}_{\mathrm{CQL}}^{\pi}$ is not a contraction. Therefore, the contraction region of $\mathcal{T}_{\mathrm{CQL}}^{\pi}$ (if exists) cannot contain the OOD region ($\beta(a|s) = 0$).

Now we consider the ID region ($\beta(a|s) > 0$).

If $\pi$ is not support-constrained, for all $(s,a)$ s.t. $\beta(a|s) > 0$ and $\forall \pi \notin \Pi$, we have

$$\begin{aligned} |\mathcal{T}_{\mathrm{CQL}}^{\pi} f_1(s,a) - \mathcal{T}_{\mathrm{CQL}}^{\pi} f_2(s,a)| &= |\mathcal{T}^{\pi} f_1(s,a) - \mathcal{T}^{\pi} f_2(s,a)| \\ &= \left| \gamma \mathbb{E}_{s' \sim P(\cdot|s,a), a' \sim \pi(\cdot|s')}[f_1(s',a') - f_2(s',a')] \right| \end{aligned} \tag{36}$$

Since $\pi \notin \Pi$, the expectation $\mathbb{E}_{a' \sim \pi(\cdot|s')}$ can have non-zero mass on some $(s',a')$ in the OOD region. Thus $\mathcal{T}_{\mathrm{CQL}}^{\pi}$ cannot be a contraction.

If $\pi$ is support-constrained, for all $(s,a)$ s.t. $\beta(a|s) > 0$ and $\forall \pi \in \Pi$, we have

$$\begin{aligned} |\mathcal{T}_{\mathrm{CQL}}^{\pi} f_1(s,a) - \mathcal{T}_{\mathrm{CQL}}^{\pi} f_2(s,a)| &= |\mathcal{T}^{\pi} f_1(s,a) - \mathcal{T}^{\pi} f_2(s,a)| \\ &= \left| \gamma \mathbb{E}_{s' \sim P(\cdot|s,a), a' \sim \pi(\cdot|s')}[f_1(s',a') - f_2(s',a')] \right| \\ &\leq \gamma \mathbb{E}_{s' \sim P(\cdot|s,a), a' \sim \pi(\cdot|s')}[|f_1(s',a') - f_2(s',a')|] \\ &\leq \gamma \max_{(s,a):\beta(a|s)>0} |f_1(s,a) - f_2(s,a)| \end{aligned} \tag{37}$$

The last inequality holds because $\pi \in \Pi$ and thus the expectation $\mathbb{E}_{a' \sim \pi(\cdot|s')}$ has non-zero mass only on support-constrained $a'$, i.e., $\beta(a'|s') > 0$.

Therefore, in the ID region ($\beta(a|s) > 0$) and when $\forall \pi \in \Pi$, $\mathcal{T}_{\mathrm{CQL}}^{\pi}$ is a $\gamma$-contraction operator. Assume the fixed point is $f^{\pi}$. We have

$$f^{\pi}(s,a) = \mathcal{T}_{\mathrm{CQL}}^{\pi} f^{\pi}(s,a) = \mathcal{T}^{\pi} f^{\pi}(s,a) - \alpha \left( \frac{\pi(a|s)}{\beta(a|s)} - 1 \right), \; \beta(a|s) > 0. \tag{38}$$

We define $R$ as the vector of reward function, and $P^\pi$ as the transition matrix on state-action pairs induced by policy $\pi$: $P^\pi_{(s,a),(s',a')} := P(s'|s,a)\pi(a'|s')$. Now Eq. (38) can be written in vector form:

$$f^\pi(s,a) = [R + \gamma P^\pi f^\pi](s,a) - \alpha \left( \frac{\pi(a|s)}{\beta(a|s)} - 1 \right), \ \beta(a|s) > 0$$

$$\Rightarrow f^\pi(s,a) = (I - \gamma P^\pi)^{-1} \left[ R - \alpha \left( \frac{\pi}{\beta} - 1 \right) \right](s,a), \ \beta(a|s) > 0$$

$$\Rightarrow f^\pi(s,a) = Q^\pi(s,a) - \alpha \left[ (I - \gamma P^\pi)^{-1} \left( \frac{\pi}{\beta} - 1 \right) \right](s,a), \ \beta(a|s) > 0$$

The row $(s,a)$ of the matrix $(I - \gamma P^\pi)^{-1}$ is $\rho^\pi_{sa}/(1-\gamma)$, i.e., the unnormalized discounted state-action occupancy induced by policy $\pi$ with initial state-action pair $(s,a)$ (Lemma 1.6 in [1]). Therefore, in the ID region ($\beta(a|s) > 0$) and when $\pi \in \Pi$, the fixed point is:

$$f^\pi(s,a) = Q^\pi(s,a) - \frac{\alpha}{1-\gamma} (\rho^\pi_{sa})^T \left( \frac{\pi}{\beta} - 1 \right), \ \beta(a|s) > 0.$$

$\square$

**Theorem 4** (Strict policy improvement to support-constrained optimal, Theorem 2). *SVR yields support-constrained $\pi_i$ and guarantees monotonic performance improvement:*

$$V^{\pi_{i+1}}(s) \geq V^{\pi_i}(s) \quad \forall s, \tag{39}$$

*where the improvement is strict in at least one state until $\pi^*_\Pi$ is found.*

*Proof.* In Theorem 3, we have proved that SVR yields support-constrained $\pi_i$ and the fixed point of policy evaluation is:

$$f^{\pi_i}(s,a) = \begin{cases} Q^{\pi_i}(s,a), & \beta(a|s) > 0, \\ Q_{\min}, & \beta(a|s) = 0. \end{cases} \tag{40}$$

Now we prove the monotonic improvement results. We start out with the performance difference lemma [16]. Given two policies $\pi', \pi$,

$$V^{\pi'}(s) - V^\pi(s) = \frac{1}{1-\gamma} \mathbb{E}_{s' \sim d^{\pi',s}}[A^\pi(s', \pi'(s'))] \tag{41}$$

where $d^{\pi',s}$ is the normalized discounted state occupancy induced by policy $\pi'$ from starting state $s$. Thus for $i > 0$,

$$V^{\pi_{i+1}}(s) - V^{\pi_i}(s) = \frac{1}{1-\gamma} \mathbb{E}_{s' \sim d^{\pi_{i+1},s}}[A^{\pi_i}(s', \pi_{i+1}(s'))] \tag{42}$$

$$= \frac{1}{1-\gamma} \mathbb{E}_{s' \sim d^{\pi_{i+1},s}}[\mathbb{E}_{a \sim \pi_{i+1}} Q^{\pi_i}(s', a) - \mathbb{E}_{a \sim \pi_i} Q^{\pi_i}(s', a)] \tag{43}$$

$$= \frac{1}{1-\gamma} \mathbb{E}_{s' \sim d^{\pi_{i+1},s}}[\mathbb{E}_{a \sim \pi_{i+1}} f^{\pi_i}(s', a) - \mathbb{E}_{a \sim \pi_i} f^{\pi_i}(s', a)] \tag{44}$$

$$\geq 0 \tag{45}$$

The third equality holds because $\pi_i$ ($i > 0$) is support-constrained and $f^{\pi_i}(s,a) = Q^{\pi_i}(s,a)$ for $\beta(a|s) > 0$ by Eq. (40). The last inequality holds because $\pi_{i+1}$ is the greedy policy with respect to $f^{\pi_i}(s,a)$ and it holds that $\mathbb{E}_{a \sim \pi_{i+1}} f^{\pi_i}(s', a) \geq \mathbb{E}_{a \sim \pi_i} f^{\pi_i}(s', a)$ at every $s'$.

When the improvement $V^{\pi_{i+1}}(s) - V^{\pi_i}(s)$ is 0 at every state $s$, it implies $\mathbb{E}_{a \sim \pi_{i+1}} f^{\pi_i}(s, a) = \mathbb{E}_{a \sim \pi_i} f^{\pi_i}(s, a)$ at every $s$, because the point mass on $s$ is "contained" in $d^{\pi_{i+1},s}$: $d^{\pi_{i+1},s}(s) > 0$.

In this case,

$$f^{\pi_i}(s,a) = \mathcal{T}^{\pi_i}_{\text{SVR}} f^{\pi_i}(s,a) = \begin{cases} r(s,a) + \gamma \mathbb{E}_{s'} \mathbb{E}_{a' \sim \pi_i(\cdot|s')}[f^{\pi_i}(s', a')], & \beta(a|s) > 0, \\ Q_{\min}, & \beta(a|s) = 0. \end{cases} \tag{46}$$

For $\beta(a|s) > 0$,

$$f^{\pi_i}(s,a) = r(s,a) + \gamma \mathbb{E}_{s'} \mathbb{E}_{a' \sim \pi_i(\cdot|s')}[f^{\pi_i}(s',a')] \tag{47}$$

$$= r(s,a) + \gamma \mathbb{E}_{s'} \mathbb{E}_{a' \sim \pi_{i+1}(\cdot|s')}[f^{\pi_i}(s',a')] \tag{48}$$

$$= r(s,a) + \gamma \mathbb{E}_{s'}\left[\max_{a' \in \mathrm{supp}(\beta(\cdot|s'))} f^{\pi_i}(s',a')\right] \tag{49}$$

The last equality holds as $\pi_{i+1}$ is support-constrained and is greedy with respect to $f^{\pi_i}$. It can be seen that Eq. (49) is the support-constrained Bellman optimality equation in Eq. (19):

Therefore,

$$f^{\pi_i}(s,a) = \begin{cases} Q_\Pi^*, & \beta(a|s) > 0, \\ Q_{\min}, & \beta(a|s) = 0. \end{cases} \tag{50}$$

Consequently,

$$\pi_{i+1}(a|s) = \mathbb{I}\left[a = \operatorname*{argmax}_{a'} f^{\pi_i}(s,a')\right] \tag{51}$$

$$= \mathbb{I}\left[a = \operatorname*{argmax}_{a' \in \mathrm{supp}(\beta(\cdot|s))} f^{\pi_i}(s,a')\right] \tag{52}$$

$$= \mathbb{I}\left[a = \operatorname*{argmax}_{a' \in \mathrm{supp}(\beta(\cdot|s))} Q_\Pi^*(s,a')\right] \tag{53}$$

$$= \pi_\Pi^* \tag{54}$$

where the last equality is the definition of the support-constrained optimal policy $\pi_\Pi^*$.

To conclude, the policy iteration with $\mathcal{T}_{\mathrm{SVR}}^\pi$ guarantees monotonic performance improvement, and the improvement is strict until the optimal support-constrained policy $\pi_\Pi^*$ is found. $\qquad\square$

## B  Experimental Details

### B.1  Experimental details on D4RL experiments

Table 2: Hyperparameters in SVR.

|  | Hyperparameter | Value |
|---|---|---|
| SVR | Optimizer | Adam [17] |
|  | Critic learning rate | $3 \times 10^{-4}$ |
|  | Actor learning rate | $3 \times 10^{-4}$ with cosine schedule |
|  | Batch size | 256 |
|  | Discount factor | 0.99 |
|  | Number of iterations | $10^6$ |
|  | Target update rate $\tau$ | 0.005 |
|  | Policy update frequency | 2 |
|  | Number of Critics | 4 |
|  | Penalty coefficient $\alpha$ | {0.001, 0.02} for Gym-MuJoCo {10} for Adroit |
|  | Standard deviation of $u$ | 0.2 |
| Architecture | Actor | input-256-256-output |
|  | Critic | input-256-256-1 |

All hyperparameters of SVR are included in Table 2. Note that the only hyperparameter we tuned is the penalty coefficient $\alpha$. We use $\alpha = 10$ for Adroit tasks and $\alpha = \{0.001, 0.02\}$ for Gym-MuJoCo tasks ($\lambda = 0.02$ for expert and medium-expert datasets, $\lambda = 0.001$ for medium, medium-replay, random datasets). The discrepancy of $\alpha$ in this two domains is due to their characteristics. Adroit

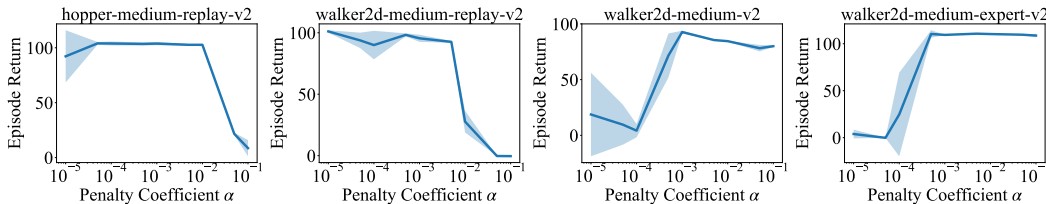

Figure 6: Parameter study on the penalty coefficient $\alpha$ of SVR. The curves are averaged over 4 random seeds, with the shaded area representing the standard deviation across seeds.

datasets are mostly collected by human behavior and the data coverage is very narrow, so strong value regularization is needed to keep value functions from overestimation. We set the standard deviation of the sampling distribution in SVR as $0.2$, which our experiments have demonstrated to be an insensitive parameter. And following TD3+BC [8], we normalize the states in all datasets.

For evaluation, we average returns over 10 evaluation trajectories and 5 random seeds on all tasks. The reported results are the normalized scores, which are offered by the D4RL benchmark [7] to measure how the learned policy compared with random and expert policy:

$$\text{D4RL score} = 100 \times \frac{\text{learned policy return} - \text{random policy return}}{\text{expert policy return} - \text{random policy return}}$$

### B.2  Experimental details on tabular maze experiments

As depicted in Fig. 1(a) in the paper, the task is to navigate from bottom-left to top-right in as few steps as possible, with a wall in the middle. The agent receives a reward of $0$ for reaching the goal and $-1$ for all other transitions. Episodes are terminated after 100 steps and $\gamma$ is set to $0.9$. We first collect $10{,}000$ transitions using a random policy. Then we remove all the transitions containing rightward actions in the bottom-left $4 \times 4$ region to introduce OOD actions. It makes the optimal support-constrained policy (see Fig. 1(b) in the paper) differ from the actual optimal policy in the environment. To magnify the impact of bootstrapping from OOD actions, we used optimistic initialization for each algorithm (i.e., initialized all $Q$-values with $Q_{\max}$). This ensures overestimation occurs in some states and we can observe how well the algorithms mitigate it.

Here we are verifying the support-constrained optimality of the SVR operator in the tabular MDP, and the actual updates of CQL and SVR are Eq. (4) and Eq. (7) in the paper respectively. We run experiments over 5 random seeds that affect dataset collection and policy/Q initialization.

### B.3  Experimental details on noisy dataset experiments

In noisy dataset experiments, we construct a "noisy" dataset by combining the random and expert datasets in D4RL with different expert ratios. The size of the combined dataset is set to $1 \times 10^6$. In some environments, the size of the random or expert dataset in D4RL is slightly smaller than $1 \times 10^6$, so we directly utilize the corresponding D4RL dataset when expert ratio is 0 or 1.

The hyperparameters of SVR follow Table 2: $\alpha = 0.001$ for expert ratio of 0 and $\alpha = 0.02$ for all other expert ratios. For CQL, we tune its regularization coefficient in $\{5, 10, 20, 30\}$ (performs relatively well in this range) and present the best results obtained for each dataset.

## C  More Experimental Results

### C.1  Ablation on the penalty coefficient

We also conduct an ablation study on the penalty coefficient $\alpha$ of SVR in Fig. 6. Experimental results indicate that different datasets have different requirements for $\alpha$, but choosing $\alpha \in [1e-3, 1e-2]$ generally induces relatively satisfying performance.

Table 3: Comparisons with additional baselines on the D4RL benchmark.

| Dataset | APAC | GAN-Joint | SPOT | MCQ | IAC | SVR (Ours) |
|---|---|---|---|---|---|---|
| halfcheetah-med | 58.4±2.6 | 44.0±0.2 | 58.4±1.0 | **64.3±0.2** | 51.6±0.3 | 60.5±1.2 |
| hopper-med | 93.8±5.4 | 86.4±10.9 | 86.0±8.7 | 78.4±4.3 | 74.6±11.5 | **103.5±0.4** |
| walker2d-med | 58.5±7.4 | 69.9±6.4 | 86.4±2.7 | 91.0±0.4 | 85.2±0.4 | **92.4±1.2** |
| halfcheetah-med-rep | **57.3±2.8** | 33.4±2.4 | 52.2±1.2 | 56.8±0.6 | 47.2±0.3 | 52.5±3.0 |
| hopper-med-rep | 51.9±5.6 | 30.9±3.2 | 100.2±1.9 | 101.6±0.8 | 103.2±1.0 | **103.7±1.3** |
| walker2d-med-rep | 16.3±5.3 | 6.7±2.2 | 91.6±2.8 | 91.3±5.7 | 93.2±1.8 | **95.6±2.5** |
| halfcheetah-med-exp | - | 72.6±11.1 | 86.9±4.3 | 87.5±1.3 | 92.9±0.7 | **94.2±2.2** |
| hopper-med-exp | - | 71.1±10.7 | 99.3±7.1 | **111.2±0.1** | 109.3±4.0 | **111.2±0.9** |
| walker2d-med-exp | - | 79.6±1.9 | 112.0±0.5 | **114.2±0.7** | 110.1±0.1 | 109.3±0.2 |
| gym total | - | 495.2 | 796.3 | 773.0 | 767.3 | **822.9** |
| pen-expert | - | 134.5±10.8 | 116.3±22.3 | - | 103.3±12.2 | **138.9±9.2** |
| pen-human | - | 71.0±23.2 | **91.4±13.4** | 68.5±6.5 | 45.9±23.0 | 73.1±12.1 |
| pen-cloned | - | 27.6±7.1 | 47.9±20.5 | 49.4±4.3 | 36.4±9.2 | **70.2±17.4** |
| adroit total | - | 233.1 | 255.6 | - | 185.6 | **282.2** |

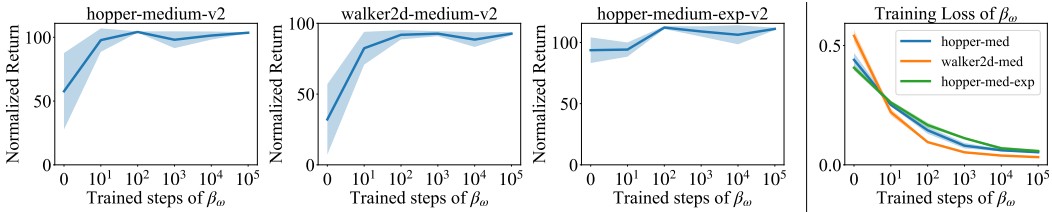

Figure 7: (Left) Performance of SVR under different behavioral model checkpoints, which are obtained at different steps in the behavioral model training process. (Right) Training loss of the behavioral model $\beta_\omega$ at different training steps. The curves are averaged over 4 random seeds, with the shaded area representing the standard deviation across seeds.

## C.2 Comparisons with additional baselines

To make a more comprehensive comparison, we also compare SVR with APAC [52], GAN-Joint [49], SPOT [43], MCQ [27], and IAC [51] on the D4RL benchmark. The results are shown in Table 3. The results indicate that our methods have better performance than these baselines.

## C.3 Empirical study on the behavior model error of SVR

To empirically investigate SVR under different behavior model errors, we run SVR using different checkpoints of the behavior model, which are obtained at different steps in the behavior model training process (Eq. (15) in the paper). The model error is controlled by the number of steps taken to train the behavioral model. The results are shown in Fig. 7. We observe that the performance of SVR increases with the number of training steps of the behavioral model. Notably, the performance of SVR stabilizes at a high level after only $10^2$ steps of behavioral model training, where the model has not been adequately trained. Theoretically, in Eq. (16) in the paper, the error of $\beta_\omega$ only affects the weights of rewarding ID actions. Thus, an imperfect model can make the maximization and minimization of ID $Q$-values not cancel out well, but have little effect on OOD ones (still penalizing all OOD $Q$-values). We hypothesize that this is the reason why SVR can still achieve good performance with an imperfect behavior model.

## C.4 Computational cost

We test the runtime of SVR on halfcheetah-medium-replay-v2 on a GeForce RTX 3090. The results of SVR and other baselines are shown in Table 4. It takes 2h40min for SVR to finish the task, which is comparable to other baselines. Note that it only takes two minutes for the pre-training part. SVR is computationally more efficient than CQL, because the practical CQL algorithm needs to sample 10

Table 4: Runtime of TD3BC, IQL, CQL, SVR for halfcheetah-medium-replay-v2 on a GeForce RTX 3090.

| Algorithm | TD3BC | IQL | CQL | SVR | pre-training in SVR |
|---|---|---|---|---|---|
| Runtime | 1h | 1h50min | 4h10min | 2h40min | 2min |

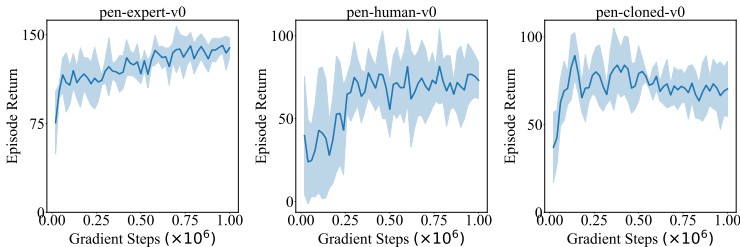

Figure 8: Learning Curves of SVR on Gym-MuJoCo Tasks.

actions at each state to compute log-sum-exp [23], while SVR only needs to sample a single action at each state. SVR can achieve better performance with lower computational costs.

### C.5 Learning curves of SVR

Learning curves on Gym-MuJoCo tasks and Adroit tasks are presented in Fig. 8 and Fig. 9, respectively. The curves are averaged over 5 random seeds, with the shaded area representing the standard deviation across seeds.

## D Broader Impact

**Societal.** Offline reinforcement learning (RL) holds significant promise in facilitating and expanding practical applications of RL, including domains such as robotics, recommendation systems, healthcare,

Figure 9: Learning Curves of SVR on Adroit Tasks.

and education, where the data collection processes are often costly or risky. However, it is important to acknowledge the potential negative societal impacts associated with any offline RL algorithm. One such concern is that the offline data used for training may contain inherent biases, and these biases can potentially transfer to the learned policy. Additionally, it is worth considering the potential impact of offline RL on employment, as it contributes to the automation of tasks traditionally performed by human experts, such as factory automation or autonomous driving. Addressing these challenges will contribute to the responsible development and deployment of offline RL algorithms, maximizing their positive impact while minimizing negative societal consequences.

**Academic.**   We reexamine the fundamental objective of value regularization in offline RL and propose supported value regularization, which not only offers enhanced theoretical guarantees but also demonstrates remarkable improvements over existing methods on widely recognized offline RL benchmarks. This research potentially offers researchers a novel perspective and a promising avenue for exploring value regularization and achieving support-constrained optimality in offline RL.

