# OpenReview forum: "Supported Value Regularization for Offline Reinforcement Learning"
_NeurIPS.cc/2023/Conference — NeurIPS 2023 poster_

### Official Review · Reviewer_2cRJ · 2023-07-02

**Soundness:** 3 good
**Presentation:** 2 fair
**Contribution:** 3 good
**Rating:** 5
**Confidence:** 4

**Summary:**

This paper studies the offline RL problem, and the authors proposes adding the Support Value  Regularization (SVR) in learning Q functions, motivated from the way how CQL add value regularizations. The authors add SVR for all OOD while maintain Bellman update for ID samples. Experiments shows that the SVR-regularized method could outperforms other competitors in D4RL tasks.

**Strengths:**

1. This paper proposes Support Value Regularization (SVR) on solving offline RL problems. The authors suggest adding extra regularization on OOD points, and utilize importance sampling techniques to calculate the Q-values in OOD region.
2. The superiority of SVR method is demonstrated from experiment studies.
3. Theoretical proof shows the policy improvement could lead to a better policy in each iteration.

**Weaknesses:**

Since the method add regularization to OOD regions, it should belongs to Conservative offline RL methods. It is suggested to compare this methods with other less/mild conservative method. In addition, as the SVR is density-based regularization, the authors is suggested to compare with other density-based Offline RL methods. Some less/mild conservative or density-based offline RL methods are listed below, the authors cites some of them, and it seems better to compared some of them.

[1] Supported Policy Optimization for Offline Reinforcement Learning.  https://arxiv.org/abs/2202.06239
[2] Mildly Conservative Q-Learning for Ofﬂine Reinforcement Learning.  https://arxiv.org/abs/2206.04745
[3] Provably Good Batch Reinforcement Learning Without Great Exploration. https://arxiv.org/abs/2007.08202
[4] A Behavior Regularized Implicit Policy for Offline Reinforcement Learning. https://arxiv.org/pdf/2202.09673.pdf
[5] APAC: Authorized Probability-controlled Actor-Critic For Offline Reinforcement Learning. https://arxiv.org/abs/2301.12130

**Questions:**

1. Some notations are not clear. In addition, what is $\beta$ in Equation 3? It also does not appear before, though we know it is behavior policy. It is suggested to give an explanation.

2. A key component in this paper is to estimate the support values. The state-action pairs are regarded as ID when $\beta(a|s) >0$, however, when using the Gaussian model to estimate behavior policy, all actions has a density larger than 0 (since gaussian density are larger than 0 everywhere), but clearly not everywhere are in distribution.

3. In addition, when estimating behavior policy $\beta_w$, the simple Gaussian model is suggested. However, as the dimension of state could be large, and the entire Support of the behavior policy relies on the behavior density estimation. A simple gaussian model may not be able to deliver a strong estimator.  In section 4.5, the authors give a simple comparison between SVR and SVR-VAE, we suggest to give more detailed analysis.

4. The sampling distribution $u(a,s)$ is selected as gaussian in Section 3.4 with mean $\pi$. Could the authors give extra abalation study on how $u(a,s)$ affects the results? Selecting $u(a,s)$ as gaussian is generally optimal/close-optimal?

**Limitations:**

See weakness and questions

---

> ### Author Rebuttal · Authors · 2023-08-09
>
> We appreciate the time and effort that you are dedicated to providing feedback on our paper and are grateful for the meaningful comments.
>
> **[W] It is suggested to compare with other less/mild conservative method. In addition, it is suggested to compare with other density-based offline RL methods.**
>
> Thanks a lot for the references. We supplement the results of these methods in Table 1 in the one-page PDF (attached to the global response). As the results show, SVR has better overall performance than these methods on both Gym-MuJoCo and Adroit tasks. Compared with less/mild conservative methods, SVR aims to solve a different, more fundamental problem. SVR focuses on which action will be penalized, instead of the strength of penalization that less/mild conservative methods considered.
>
> **[Q1] Some notations are not clear, e.g.,  $\beta$ in Equation 3.**
>
> Sorry for that. Before Equation 3, $\beta$ appears in Line 75 "a fixed dataset $\mathcal{D}$ collected by some behavior policy $\beta$". We will make it clearer in the latter revision.
>
> **[Q2,Q3] The state-action pairs are regarded as ID when $\beta(a|s)>0$. However, when using the Gaussian model to estimate behavior policy, all actions have a density larger than 0, but clearly not everywhere is in distribution. As the dimension of state could be large, and the entire support of the behavior policy relies on the behavior density estimation. A simple Gaussian model may not be able to deliver a strong estimator.**
>
> Thanks for your comment. The definition of ID state-action pairs is $\beta(a|s)>0$ where $\beta$ is the true behavior policy. However, with an estimated behavior policy $\hat{\beta}$, SVR does not distinguish between ID and OOD actions by whether $\hat{\beta}>0$, as the density of OOD actions is particularly difficult to estimate and the behavioral model error can cause extrapolation error and overestimation in this way. Instead, like Eq.5, SVR penalizes all the actions in action space and compensates for ID actions. When $\hat{\beta}$ is estimated accurately, the regularization effects on ID Q-values cancel out, achieving penalization for all OOD Q-values only (Eq.6). **With model error of $\hat{\beta}$ incorporated, $\mathbb{E}_{a \sim \beta}$ in Eq.5 is not affected and the model error only affects the weight of rewarding true ID actions (the IS ratio in the last term)**. As a result, an imperfect model can make the minimization and maximization for ID Q-values not cancel out well, but have little effect on OOD ones (still penalizing all OOD Q-values). **In conclusion, the behavioral model is not used to distinguish between ID and OOD actions in SVR, and the model error can only lead to some unnecessary changes to ID Q-values and will not generate extrapolation error or OOD overestimation.** In addition, compared with other methods that require the behavior model, SVR is less susceptible to model errors because SVR only needs to query the behavior density of in-dataset (s, a) pairs, thus not requiring much generalization ability of the model, making it relatively easier to estimate accurately.
>
> To empirically investigate SVR under different behavioral model errors, we run SVR using different checkpoints of the behavioral model, which are obtained at different steps in the behavioral model training process. The model error is controlled by the number of steps taken to train the behavioral model. The results are shown in Figure 2 in the one-page PDF. We observe that the performance of SVR increases with the number of training steps of the behavioral model. Notably, the performance of SVR stabilizes at a high level after only 1e2 steps of behavioral model training, where the model has not been adequately trained. It indicates that SVR can achieve good performance with an imperfect behavioral model.
>
>
> **[Q3] In section 4.5, the authors give a simple comparison between SVR and SVR-VAE, we suggest to give more detailed analysis.**
>
> Thanks for your suggestion. In Section 4.5, we found that the advantages of the VAE estimator are not well demonstrated under the common D4RL datasets. Thus, we conduct additional experiments on bimodal datasets, which are constructed by mixing hopper-expert dataset and another dataset collected by a narrow and highly suboptimal Gaussian policy $N(0,0.04)$. In this case, Gaussian can not model the behavior policy well. The results are shown in Figure 3 in the one-page PDF. Benefit from the flexibility of VAE estimator, SVR-VAE obtains better results.
>
>
> **[Q4] The sampling distribution $u(a|s)$ is selected as gaussian in Section 3.4 with mean $\pi$. Could the authors give extra abalation study on how $u(a|s)$ affects the results? Selecting $u(a|s)$ as gaussian is generally optimal/close-optimal?**
>
> Thanks for your comment. We conducted this ablation in Section 4.4 of the paper, including Guassian with various variance and the uniform distribution. It is shown that SVR is insensitive to $u$ in a wide range. Theoretically, a choice of $u$ is optimal as long as it covers the entire action space. Empirically, Guassian with moderate variance satisfies this requirement and can emphasize the areas where overestimation is most likely to occur (as indicated by the current policy). We only considered Gaussian and uniform distributions as they are the most common samplable continuous distributions. While it is possible to design more complex $u$ (like based on uncertainty estimation), it can be prohibitively difficult or expensive to sample from such complex distributions.

---

> > ### Comment · Reviewer_2cRJ · 2023-08-19
> > **Thank you for your response**
> >
> > Thank you for your reply. Your explanation addresses some of my concerns, and I appreciate you for conducting extra experiments.

---

> > > ### Author Response · Authors · 2023-08-19
> > > **Thank you!**
> > >
> > > Thank you for your feedback and dedication to our paper!
> > >
> > > We are happy to have addressed some of your concerns. May we kindly ask if you still have concerns or questions unaddressed? If so, we really want to discuss and address them in the time we have. If our response and additional experiments have addressed your major concerns, would you mind considering increasing your score based on the updated information? Following the valuable suggestions from you and other reviewers, we believe this work has been further strengthened.
> > >
> > > Thanks!

---

### Official Review · Reviewer_c3nj · 2023-07-04

**Soundness:** 2 fair
**Presentation:** 3 good
**Contribution:** 2 fair
**Rating:** 5
**Confidence:** 4

**Summary:**

This paper proposes the use of Importance sampling to distinguish between ID and OOD actions, and to operate the corresponding actions accordingly. In addition, as most of the current work is similar, it also uses model to fit behavior policy, but does not need to overly consider the accuracy of the model. It underestimates OOD actions and unbiased estimates ID actions, achieving good performance on D4RL and Adroit tasks.

**Strengths:**

1. The paper conducts relatively sufficient experiments to demonstrate its viewpoints.
2. The proofs of theorems and propositions are relatively sufficient.
3. The hyperparameters and parameters settings in the paper are clearly defined for easy reproduction.


**Weaknesses:**

1. Importance sampling is widely used in reinforcement learning because of its unbiased nature. In this paper, the main contribution is to use importance sampling to punish the deviation of OOD action and ID action respectively. The differences or advantages between the proposed SVR method and existing methods should be described clearly.
2. The description of certain experiments should be clearer. For example, what do the colors of the grid represents in Figure 1? In Figures 1 and 2, it is hard to see how the value function described in the paper is represented.
3. The algorithms in the comparative experiment are only some classic algorithms, lacking some algorithms of the same type, such as using importance resampling (Zhang et al. 2023), and there is no mention of the ablation experiments about the penalty coefficient α.
(Zhang et al. 2023) Hongchang Zhang, Yixiu Mao, Boyuan Wang, Shuncheng He, Yi Xu, Xiangyang Ji. In-sample Actor Critic for Offline Reinforcement Learning. International Conference on Learning Representations, 2023.
https://openreview.net/forum?id=dfDv0WU853R



**Questions:**

1. Recently, there has been a way to use Importance sampling or Importance resampling to determine whether it belongs to OOD actions. Please explain the relationship between your work and the current works, or explain the differences and advantages of your method.
2. In Figure 2, I cannot see that there has a serious overestimation problem with the value function caused by the iteration of the vanilla policy.
3. Please provide a more detailed explanation of Figure 1, such as what the colors of the grid represent.
4. Why does the Walker2d-v2-random in Figure 1 in appendix only run a small portion and not continue to run?


**Limitations:**

There is no negative societal impact.

---

> ### Author Rebuttal · Authors · 2023-08-09
>
> We appreciate the time and effort that you are dedicated to providing feedback on our paper and are grateful for the meaningful comments.
>
> **[W1, Q1] Importance sampling is widely used in reinforcement learning because of its unbiased nature. The differences or advantages between the proposed SVR method and existing methods should be described clearly. Recently, there has been a way to use Importance sampling or Importance resampling to determine whether it belongs to OOD actions.**
>
> Thanks for your suggestion. Importance sampling's application in RL has a long history. Most of them utilize some form of importance sampling to either evaluate the return of a given policy, or to estimate the corresponding policy gradient [1,2]. However, they can have very high variance due to the product of importance weights. In contrast, SVR focuses on value regularization in offline RL and involves only one importance weight. Recently in offline RL, some works use importance sampling (or importance resampling) to realize in-sample learning [3,4]. They formulate the Bellman target with the actions in the dataset (SARSA update), and weigh the update by an importance ratio. Thus, they do not differentiate between ID and OOD actions or perform regularization on Q functions. One limitation of these methods is that, they only deal with in-sample actions (actions in the dataset) and can not take advantage of the generalization ability of neural networks. The goal of SVR is quite different from these works - to penalize all OOD Q-values without affecting ID ones. To the best of our knowledge, our work is the first to leverage importance sampling to achieve proper value regularization in offline RL.
>
> **[W2, Q3] What do the colors of the grid represent in Figure 1? In Figures 1 and 2, it is hard to see how the value function described in the paper is represented.**
>
> Sorry for the unclear description. The colors of the grid in Figure 1 represent the value functions of the corresponding states, and their specific values are indicated by the color bar on the right side of Figure 1. In addition, the gray/white rectangle area in the middle of the map represents a wall.
>
> **[W3] Lack some algorithms of the same type in the comparative experiment, such as using importance resampling [4], and there is no mention of the ablation experiments about the penalty coefficient $\alpha$.**
>
> Thanks lot for your suggestions. We supplement the results of IAC [4] in Table 1 in the one-page PDF (attached to the global response) for comparison. It shows that SVR has better performance than IAC on both Gym-MuJoCo and Adroit tasks. In addition, we also conduct an ablation study on the penalty coefficient $\alpha$ in Figure 1 in the PDF. Experimental results indicate that choosing  $\alpha \in [1e-3, 1e-2]$ generally induces good performance.
>
> **[Q2] In Figure 2, I cannot see that there is a serious overestimation problem with the value function caused by the iteration of the vanilla policy.**
>
> Sorry for the unclear description. Figure 2 presents the Episode Return obtained by evaluating the learned policy at each iteration, which represents the actual performance of learned policy, rather than learned value function. As mentioned in [W2, Q3], the colors of the grid in Figure 1 represent the learned value functions. It shows that vanilla policy iteration has a severe overestimation of value functions.
>
> **[Q4] Why does the Walker2d-v2-random in Figure 1 in appendix only run a small portion and not continue to run?**
>
> This is because a severe overestimation occurred and we stopped running automatically. Walker2d-random has extremely narrow data coverage and most of the existing algorithms suffer from severe over-estimation, resulting in poor performance (see Table 1 in the paper). While choosing a large penalty coefficient $\alpha$ in SVR can mitigate the over-estimation in Walker2d-random, we don't set a particular hyperparameter for it, as hyperparameter tuning in offline RL should be avoided as much as possible.
>
> [1] Doina Precup. Eligibility traces for off-policy policy evaluation. Computer Science Department Faculty Publication Series, 2000.
>
> [2] Nan Jiang and Lihong Li. Doubly robust off-policy value evaluation for reinforcement learning. International Conference on Machine Learning,  2016.
>
> [3] Yiqin Yang, et al. Believe what you see: Implicit constraint approach for ofﬂine multi-agent reinforcement learning. Advances in Neural Information Processing Systems, 2021.
>
> [4] Hongchang Zhang, et al. In-sample Actor Critic for Offline Reinforcement Learning. International Conference on Learning Representations, 2022.

---

> > ### Comment · Reviewer_c3nj · 2023-08-17
> >
> > Thank the authors for their effort in addressing the concerns and providing the additional experiment results.
> > By replying to W1 and Q1, we can conclude that the main difference between SVR and existing works is that it utilizes the generalization ability of neural networks rather than conducting in-sample learning. However, this approach is similar to conservative learning methods, which penalizes OOD actions without affecting ID actions.
> > For others, the authors provide more detailed descriptions of the figures and the paper.
> > After carefully considering the others reviews and the corresponding rebuttals, we decide to maintain the original score.

---

> > > ### Author Response · Authors · 2023-08-17
> > > **Thank you for your feedback**
> > >
> > > Thank you for your feedback! We are happy to address some of your questions, but we still want to clarify a few things.
> > >
> > > Utilizing the generalization ability of neural network is an advantage of conservative methods over in-sample methods. SVR belongs to conservative methods. **The main contribution of SVR is that it penalizes *all* OOD actions *without* affecting ID actions, which, to our knowledge, is *not* realized by other conservative methods even though they aim to do so**. Benefit from this property, SVR guarantees optimal convergence in tabular MDP and shows impressive performance empirically. Importance sampling is just a component in SVR to realize this idea.

---

### Official Review · Reviewer_DikP · 2023-07-25

**Soundness:** 3 good
**Presentation:** 3 good
**Contribution:** 2 fair
**Rating:** 5
**Confidence:** 4

**Summary:**

This paper proposes a new offline RL method in which the popular  assumption that the new policy should be close to the behavior poplicy is abandoned and just penalizing the OOD action would be OK. Analysis show that the method has the policy improvement property. Experimental results partially verify the effectiveness of this 'more simple' method (compared to CQL).

**Strengths:**

The presentatation is clear and the proposed method is easy to understand.

The final results look good.

**Weaknesses:**

1. By relaxing the behavior clone requirement (first eq of eq(4)) and only penalizing the OOD action, the new policy would have better chance to find a better one. This is reasonable. However, since the search space becomes larger, the learning process should become more difficult. Unfornuately, the experimental results (Table 1) are eager to show this method is good, without illustrating how the reward is accumulated  as usually done.

2. According to eq.(7), this method would be even more sensitive to the quality of the learnt behavior policy itself than CQL, while the behavior policy  is generlly quite difficult to learn. This may generate extrapolation error. For example, if we utilize a Gaussian approximal behavior policy and the a1, a2 are both the in-sample actions at state s. Then the intropolated OOD a3 between a1 and a2 would likely has a large probability density of the behavior policy, which would introduce extropolation error. Hence the paper should provide more evidence to  support the claim that 'SVR is less susceptible to model errors' .

3. Generalization problem. The proposed SVR could be considered as a clipped Conservative Q-Learning, where the part with ID actions is dropped. Then the Q value generated by this method would have the same generalization problem as CQL. For example, if we have learnt an approximal behavior policy perfectly, then the Q value of any perturbed state-action pair (unseen but closed to the dataset) would drop greatly, although we may need a smoother Q function to enhance the robustness [RORL: Robust Offline Reinforcement Learning via Conservative Smoothing].

4. Lack of the experiments that exploring the behavior of the SVR agent. That is, the paper is supposed to show how the SVR behaves at the ID states and OOD states in practice.

5.  Although the theoretical analysis demonstrates the performance improvement guarantee of SVR, it deviates from the core problem that this paper attempts to solve, where SVR introduces smaller detrimental changes to in-distribution Q-values.


**Questions:**

see above

---

> ### Author Rebuttal · Authors · 2023-08-08
>
> We appreciate the time and effort that you are dedicated to providing feedback on our paper and are grateful for the meaningful comments.
>
> **[W1] The experimental results (Table 1) are eager to show this method is good, without illustrating how the reward is accumulated as usually done.**
>
> We included all the learning curves of SVR in Section B of the appendix. Overall, SVR has a stable learning process and good asymptotic performance.
>
> **[W2] According to eq.(7), this method would be even more sensitive to the quality of the learnt behavior policy. It may generate extrapolation error. The paper should provide more evidence to support the claim that 'SVR is less susceptible to model errors'.**
>
> Thanks for your suggestion. With an estimated behavior policy $\hat{\beta}$, SVR does not distinguish between ID and OOD actions by whether $\hat{\beta}>0$ like Eq.7, as the density of OOD actions is particularly difficult to estimate and the behavioral model error can generate extrapolation error in this way as you mentioned. Instead, like Eq.5, SVR penalizes all the actions in action space and compensates for ID actions. When $\hat{\beta}$ is estimated accurately, the regularization effects on ID Q-values cancel out, achieving penalization for all OOD Q-values only (Eq.6), leading to the optimal solution Eq.7. However, **with model error of $\hat{\beta}$ incorporated, Eq.7 does not become that where $\beta$ is substituted by $\hat{\beta}$**, because in Eq.5, the model error does not affect $\mathbb{E}_{a \sim \beta}$ and only affects the weight of rewarding ID actions (the IS ratio in the last term). As a result, an imperfect model can make the minimization and maximization for ID Q-values not cancel out well, but have little effect on OOD ones (still penalizing all OOD Q-values). **In conclusion, the behavioral model is not used to distinguish between ID and OOD actions in SVR, and the model error can only lead to some unnecessary changes to ID Q-values and will not generate extrapolation error or OOD overestimation.** In addition, compared with other methods that require the behavior model, SVR is less susceptible to model errors because SVR only needs to query the behavior density of in-dataset (s, a) pairs, thus not requiring much generalization ability of the model, making it relatively easier to estimate accurately.
>
> To empirically investigate SVR under different behavioral model errors, we run SVR using different checkpoints of the behavioral model, which are obtained at different steps in the behavioral model training process. The model error is controlled by the number of steps taken to train the behavioral model. The results are shown in Figure 2 in the one-page PDF (attached to the global response). We observe that the performance of SVR increases with the number of training steps of the behavioral model. Notably, the performance of SVR stabilizes at a high level after only 1e2 steps of behavioral model training, where the model has not been adequately trained. It indicates that SVR can achieve good performance with an imperfect behavioral model.
>
> **[W3] The Q value generated by this method would have the same generalization problem as CQL.**
>
> Thanks for your meaningful comment. Recently, some works consider the generalization problem of Q functions, aiming to learn smoother or less conservative Q functions [1,2]. In contrast, SVR aims to solve a different, more fundamental problem. SVR focuses on which action will be penalized, instead of the strength of penalization that these methods consider. In addition, these methods that improve the generalization of Q-networks can also be combined into SVR in practice. For example, substitute $Q_{\mathrm{min}}$ in Eq.16 with a value that is slightly smaller than in-distribution maximum, as MCQ did [2].
>
> [1] Rui Yang, et al. RORL: Robust Offline Reinforcement Learning via Conservative Smoothing. Advances in Neural Information Processing Systems, 2022.
>
> [2] Jiafei Lyu, et al. Mildly Conservative Q-Learning for Ofﬂine Reinforcement Learning. Advances in Neural Information Processing Systems, 2022.
>
> **[W4] The paper is supposed to show how the SVR behaves at the ID states and OOD states in practice.**
>
> Thanks for your meaningful comment. The main purpose of this work is to punish OOD actions, which causes value over-estimation. Compared to OOD actions, OOD states cause less severe problems (mainly the state deviation issue during test time [3]), and much fewer works focus on it. Since the contribution of SVR does not lie in handling OOD states, we only investigate the behavior of SVR with respect to ID/OOD actions. Figure 1 in the paper shows that SVR not only avoids taking OOD actions, but also chooses the optimal ID actions.
>
> [3] Hongchang Zhang, et al. State Deviation Correction for Offline Reinforcement Learning. Proceedings of the AAAI Conference on Artificial Intelligence, 2022.
>
> **[W5] Although the theoretical analysis demonstrates the performance improvement guarantee of SVR, it deviates from the core problem, where SVR introduces smaller detrimental changes to in-distribution Q-values.**
>
> We would like to make an emphasis on Theorem 1. According to it, the policy iteration with SVR operator outputs unbiased Q-values for all ID actions and underestimated Q-values for all OOD actions. Benefit from this property, SVR can achieve performance improvement and optimal convergence guarantee.

---

> > ### Comment · Reviewer_DikP · 2023-08-17
> >
> > Thanks for the clarification and I would keep my previous score unchanged.

---

> > > ### Author Response · Authors · 2023-08-17
> > > **Thank you for your feedback**
> > >
> > > Thank you for your feedback! If you have any further questions, please post them. We would be more than happy to resolve any remaining questions in the time we have, and are looking forward to engaging in a discussion.

---

### Official Review · Reviewer_CX3L · 2023-07-25

**Soundness:** 2 fair
**Presentation:** 3 good
**Contribution:** 2 fair
**Rating:** 5
**Confidence:** 3

**Summary:**

The authors propose to enforce a new squared penalization term for computing  the target Q-values in offline RL. In particular, their penalty tries to only apply to target out-of-distribution (OOD) actions by taking the difference between the importance-sampled and the true estimates of a uniform distribution over actions (or any other distribution will full support on the action space), where the importance-ratio is estimated with a proxy model for the behavior policy. Under idealized theoretical settings, the paper shows that off-policy optimization using this penalty has a fixed point which penalizes only OOD actions and guarantees strict policy improvements. The authors empirically validate their algorithm showing effective performance on a subset of D4RL and a toy maze setting.

**Strengths:**

1) The paper provides a concise and comprehensive introduction to the off-policy literature and the utilized notation in Section 2.
2) The paper correctly recognizes an important issue of existing popular algorithms in the offline literature: while their stated aim is to penalize behavior only outside the support set of the offline datasets, in practice they resort to density-based heuristic penalties.
3) Overall, the paper is clear and easy to read.
4) The authors provide some analysis and ablations beyond just reporting their method's performance.

**Weaknesses:**

1) Generalizing some of the mentioned regularization issues to the whole literature would require very comprehensive evidence which the paper does not provide,  as it mostly focuses on CQL when analyzing prior work. Hence, I would tone down some of the statements, e.g. lines 26-28"existing value regularization methods not only fall short in penalizing all OOD Q-values but also may introduce detrimental changes to in-distribution (ID) ones" -> " some of the most popular existing value...". Additionally, claims like "SVR guarantees strict policy improvement until convergence to the optimal support-constrained policy" (333-334) should specify that this assumes an idealized setting, and does not apply in the stochastic optimization regime of deep learning.
2) Related to the point above, I believe it would be very appropriate for the paper to mention the assumptions/conditions used to derive the theoretical results in the introduction and abstract to avoid having unbacked claims.
3) A limitation of the method is that it requires access to the behavior's policy density.  It would be useful to also analyze the properties of the proposed penalization (theoretically and/or empirically) based on the modeling errors for the trained behavior policy proxy.
4) In practice, CQL's policy is most often also a Gaussian distribution, and, thus, it covers the full support of the action space. Yet, the authors do not seem to consider or mention this property in their analysis. Moreover, since the authors also resolve to using the agent's Gaussian policy with an increased variance for the sampling (action-space covering) penalization distribution, it appears to me that the actual implementation of their algorithm is very similar to existing off-policy methods (e.g., even to CQL).
5) The toy example in Section 4.1 seems quite misleading. From a practical perspective, it seems to me like CQL and SVR are algorithms with similar properties whose performance depends on hyperparameters.  Hence, I believe that simply showing one works and the other fails with no context (swept hyperparameters/different collected datasets) is not very informative.
6) While the authors show some extensions of SVR replacing the Gaussian density estimator with a conditional VAE and using self-normalized importance sampling, results appear inconclusive and there is very little analysis done to motivate these findings.
8) Selection of the distribution to cover the whole action space, clearly influences the penalization magnitude of different actions (Equations 5-6), and, therefore, performance.  There does not appear to be be any clear principle for choosing this important hyper-parameter for a given problem - apart from looking at online performance, which would break the assumptions of a fully-offline setting.

Minor:
Typos - Line 108: 'in policy improvement stage' -> 'in the policy improvement stage', Line 108: scare -> scarce.

**Questions:**

- Since also CQL uses a Gaussian policy with full support over the action space, do some of the theoretical claims of SVD generalize beyond this specific algorithm?
- What is the intuition and what are the theoretical consequences of applying SNIS over importance sampling (Section 4.5)?
- Even if considered a standard benchmark in the offline literature, I generally do not find evaluating on a subset of D4RL very informative in terms of empirically properties. Have the authors considered evaluating on additional/more comprehensive offline benchmarks? (e.g. [1])

[1] Gulcehre, Caglar, et al. "Rl unplugged: A suite of benchmarks for offline reinforcement learning." Advances in Neural Information Processing Systems 33 (2020).

**Limitations:**

The authors quickly dismiss the constraint of having to model the behavior policy by stating that "we empirically find that the Gaussian model can usually induce excellent performance, which only takes two minutes for pre-training." (Lines 337-338) This claim should be contextualized, as a shallow Gaussian policy is most definitely not sufficient beyond the considered toy settings and the relatively simple D4RL benchmark.

---

> ### Author Rebuttal · Authors · 2023-08-09
>
> We appreciate the time and effort that you are dedicated to providing feedback on our paper and are grateful for the meaningful comments.
>
> **[W1] I would tone down some of the statements, e.g. lines 26-28"existing value regularization methods" -> " some of the most popular existing value...".**
>
> Thanks for your constructive comment. We used "existing value regularization methods" because we tried our best to summarize the existing value regularization algorithms in the related work section, and to the best of our knowledge, few works achieve the original purpose of value regularization - to penalize all OOD Q-values without affecting ID ones.
>
> **[W1, W2] Mention the assumptions/conditions used to derive the theoretical results in the introduction and abstract.**
>
> Thanks a lot for pointing it out. We analyze the policy iteration with the SVR operator in the tabular MDP setting, which is common in the analyses of offline RL algorithms [1,2]. We will make sure to add it in the latter revision.
>
> [1] Seyed Kamyar Seyed Ghasemipour, et al. EMaQ: Expected-Max Q-Learning Operator for Simple Yet Effective Ofﬂine and Online RL. ICML, 2021.
>
> [2] Jiafei Lyu, et al. Mildly Conservative Q-Learning for Ofﬂine Reinforcement Learning. NeurIPS, 2022.
>
> **[W3] It would be useful to also analyze the properties of the proposed penalization (theoretically and/or empirically) based on the modeling errors for the trained behavior policy proxy.**
>
> Thank you for your valuable suggestion. To empirically investigate SVR under different behavioral model errors, we run SVR using different checkpoints of the behavioral model, which are obtained at different steps in the behavioral model training process. The model error is controlled by the number of steps taken to train the behavioral model. The results are shown in Figure 2 in the one-page PDF (attached to the global response). We observe that the performance of SVR increases with the number of training steps of the behavioral model. Notably, the performance of SVR stabilizes at a high level after only 1e2 steps of behavioral model training, where the model has not been adequately trained.
>
> Theoretically, in SVR, the error of the behavioral model $\beta_\omega$ only affects the weights of rewarding ID actions (see Eq. 16). Thus, an imperfect model can make the maximization and minimization of ID Q-values not cancel out well, but have little effect on OOD ones (still penalizing all OOD Q-values). We hypothesize that this is the reason why SVR can achieve good performance with an imperfect behavioral model.
>
> **[W4, Q1] Since CQL also uses a Gaussian policy with full support over the action space, do some of the theoretical claims of SVR generalize beyond this specific algorithm?**
>
> In our analyses of CQL, the policy $\pi$ can be any distribution, including Gaussian covering the full action space. According to Proposition 3, $\mathcal{T}^\pi_{CQL}$ is a contraction operator only if $supp(\pi) \subseteq supp(\beta)$. This is consistent with the analysis in CQL's paper, which also assumes this strong assumption. On the one hand, Gaussian policy covering the full action space does not satisfy this assumption of CQL. On the other hand, even for $\pi$ satisfying the assumption, the fixed point may underestimate or overestimate Q-values in a complicated way (Proposition 3). Therefore, the theoretical claims of SVR cannot be extended to CQL. In addition, from a practical perspective, the Guassian policy in CQL is usually very narrow and hardly covers the action space.
>
> **[W4] It appears to me that the actual implementation of their algorithm is very similar to existing off-policy methods (e.g., even to CQL).**
>
> Yes, our implementation is similar to CQL but with two main differences: much wider penalization for all Q-values and weighted maximization for ID Q-values.
>
> **[W5] In the toy example, I believe that simply showing one works and the other fails with no context (swept hyperparameters/different collected datasets) is not very informative.**
>
> Sorry that we miss some experimental details. In the toy example, we were verifying the support-constrained optimality of the SVR operator in the tabular MDP, and the actual updates of CQL and SVR are like Eq.4 and Eq.7 respectively, which are quite different. We ran experiments over 5 random seeds that affect dataset collection and policy/Q initialization, and the learning curves are shown in Figure 2. We also swept the hyperparameter, but the results were similar (SVR converged, CQL did not). We will add the details in the latter revision.
>
> **[W6, Q2] While the authors show some extensions of SVR replacing the Gaussian density estimator with a conditional VAE and using self-normalized importance sampling, results appear inconclusive and there is very little analysis done to motivate these findings. What is the intuition and what are the theoretical consequences of applying SNIS over IS?**
>
> SVR-VAE: We find that the advantages of VAE are not well demonstrated under the common D4RL datasets. Thus, we conduct additional experiments on bimodal datasets, which are constructed by mixing hopper-expert dataset and another dataset collected by a narrow and highly suboptimal Gaussian policy $N(0,0.04)$. In this case, Gaussian can not model the behavior policy well. The results are shown in Figure 3 in the one-page PDF. Benefit from the flexibility of VAE estimator, SVR-VAE obtains better results.
>
> SVR-SNIS: Theoretically, SNIS is biased, but the bias is small, and the improvement in variance makes it a preferred alternative to IS sometimes [3]. The results in Section 4.5 show that SVR-SNIS performs comparably to SVR in most tasks but worse than SVR on hopper-med, probably due to the bias.
>
> [3] Art B. Owen. Monte Carlo theory, methods and examples. 2013.
>
> **Due to the page limit, please refer to the global response block on the top of this page for the remaining responses. Thanks!**

---

> > ### Comment · Reviewer_CX3L · 2023-08-17
> > **Thank you for your rebuttal**
> >
> > I thank the authors for acknowledging and responding to my main concerns, and I hope they will include the promised changes in future revisions. After also reading the other reviews, my overall assessment of the paper is still positive, although I do not have a very strong opinion given the incremental nature of the work and the modifications required from the submitted version.

---

> > > ### Author Response · Authors · 2023-08-18
> > > **Thank you for your feedback**
> > >
> > > Thank you for your feedback! We really appreciate your suggested modifications to make the statements more rigorous. However, we still want to make a few clarifications.
> > >
> > > We agree with you that "The paper correctly recognizes an important issue of existing popular algorithms in the offline literature". The main contribution of SVR is that it penalizes *all* OOD Q-values *without* affecting ID ones, which, to our knowledge, is not realized by other value regularization methods (conservative methods) even though they aim to do so. Benefit from this property, SVR guarantees support-constrained optimal convergence in tabular MDP and outperforms prior methods by a large margin. Thus we believe both in theory and in practice, SVR is not an incremental work and may indicate a direction for subsequent value regularization works in offline RL.
> > >
> > > Following your suggestions, we believe that we have made a great effort to provide all the experiments that we can (behavioral model error, bimodal datasets, RL unplugged benchmark), and the results further demonstrate the superiority of SVR. We sincerely appreciate it if you could re-evaluate the contribution of this work. We will make the contributions of this work clearer in the revision. Thank you very much for your time and efforts.

---

### Author Rebuttal · Authors · 2023-08-09

### **Global Response**

We thank all the reviewers for the insightful comments and suggestions. We are greatly encouraged by the positive comments of reviewers, e.g.,

* The paper correctly recognizes an important issue of existing popular algorithms in the offline literature. (CX3L)
* The presentation is clear and the proposed method is easy to understand. (DikP)
* The proofs of theorems and propositions are relatively sufficient. (c3nj)
* The superiority of SVR method is demonstrated from experiment studies. (2cRj)

Meanwhile, we have made every effort to address all the reviewers' concerns and responded to the individual reviews below. We have also uploaded a **one-page PDF** (attached to this response) that contains the additional experiment results. Summary of the PDF:

* Comparisons with additional baselines on the D4RL benchmark in Table 1.
* Experimental results on a subset of the RL unplugged benchmark in Table 2.
* Ablation results on the penalty coefficient $\alpha$ of SVR in Figure 1.
* Experimental results of SVR under different behavioral model errors in Figure 2.
* Comparisons between SVR-VAE and SVR-Gaussian on the bimodal datasets in Figure 3.

We hope our response could address the reviewers' concerns. We would be more than happy to resolve any remaining questions in the time we have, and are looking forward to engaging in a discussion.

---

### **Additional Response to Reviewer CX3L (Part 2 of 2)**

**[W7] There does not appear to be any clear principle for choosing this important hyper-parameter (the sampling distribution $u$) for a given problem - apart from looking at online performance.**

We conducted a parameter study on the sampling distribution $u$ in Section 4.4. As shown in Figure 4, SVR is insensitive to $u$ in a wide range. We also propose an intuitive approach for choosing $u$ - Gaussian with the same mean as the current policy to emphasize the areas where overestimation is most likely to occur and with a moderate variance to cover the entire action space.

**[Q3] Have the authors considered evaluating on additional/more comprehensive offline benchmarks? (e.g. Rl unplugged)**

Thank you for your suggestion. The experimental results on a subset of the RL unplugged benchmark are shown in Table 2 in the one-page PDF. We observe that SVR performs better than baseline methods on three tasks and slightly worse on one task.

**[L1] (Lines 337-338) This claim should be contextualized, as a shallow Gaussian policy is most definitely not sufficient beyond the considered toy settings and the relatively simple D4RL benchmark.**

We apologize for the unclear statement. While the Gaussian behavioral model performs well empirically in D4RL, it can be intuitively problematic when dealing with datasets with complex distributions. As mentioned in [W6, Q2], we conduct additional experiments on the bimodal datasets, where SVR-VAE outperforms SVR due to the flexibility of the VAE estimator.

---

### Decision · Program_Chairs · 2023-09-21

**Decision:**

Accept (poster)

**Comment:**

I went through the reviews and discussion carefully, as well as through the paper. Although this work builds on CQL, it is a fundamentally distinct and reasonably novel approach. I thought the authors' analysis in sections 3.1 and 3.2 motivated things very well, and to me it seems like an interesting enough idea that others can build on.

The fact that this is complemented with theoretical proofs of convergence and improvement, as well as strong empirical performance, leads me to recommend acceptance of this paper.